



**Evolution of the Iberian Massif as deduced from its crustal thickness and**
**geometry of a mid-crustal (Conrad) discontinuity**
**Puy Ayarza[1], José Ramón Martínez Catalán[1], Ana Martínez García[2], Juan Alcalde[2], Juvenal Andrés[1,2],**
**José Fernando Simancas[3], Immaculada Palomeras[1], David Martí[2,4], Irene DeFelipe[2], Chris Juhlin[5],**
**Ramón Carbonell[2]**
1. Geology Department, Salamanca University. Pza de la Merced, s/n. Salamanca 37008. Spain
2. Geosciences Barcelona, CSIC. Lluis Solé i Sabaris, s/n. Barcelona 08028. Spain
3. Geodynamics Department, Granada University. Av/ de la Fuente Nueva, s/n. Granada 18071. Spain
4. Lithica SCCL. Avinguda Farners 16, Santa Coloma de Farners, Girona 17430. Spain
5. Department of Earth Sciences, Uppsala University. Villavägen 16, Uppsala, 75236. Sweden
Corresponding author: Puy Ayarza (puy@usal.es)
**Abstract**
Normal incidence seismic data provide the best images of the crust and lithosphere. When
properly designed and continuous, these sections greatly contribute to understanding the
geometry of orogens and, together with surface geology, to unravel their evolution. In this
paper we present an almost complete transect of the Iberian Massif, the westernmost
exposure of the European Variscides. Despite the heterogeneity of the dataset, acquired
during the last 30 years, the images resulting from reprocessing with a homogeneous workflow
allow us to clearly define the crustal thickness and its internal architecture. The Iberian Massif
crust, formed by the amalgamation of continental pieces belonging to Gondwana and
Laurussia (Avalonian margin) is well structured in upper and lower crust. A conspicuous mid-
crustal discontinuity is clearly defined by the top of the reflective lower crust and by the
asymptotic geometry of reflections that merge into it, suggesting that it has often acted as a
detachment. The geometry and position of this discontinuity can give us insights on the
evolution of the orogen, i.e. of the effects and extent of the late Variscan gravitational
collapse. Also, its position and the limited thickness of the lower crust in central and NW Iberia
constraints the response of the Iberian microplate to Alpine shortening. This discontinuity is
here observed as an orogeny-scale feature with characteristics compatible with those of the
worldwide, Conrad discontinuity.
**Keywords**: Iberian Massif, vertical incidence seismic data, mid-crustal detachment, Conrad
discontinuity, geodynamic evolution
**1. Introduction**
In the last 35 years, controlled source seismic data have greatly contributed to the
understanding of the European Variscides. National research programs like DEKORP (Bortfeld,
1985; DEKORP Research Group, 1987; Franke et al., 1990; Oncken, 1998), BIRPS and ECORS
(BIRPS and ECORS, 1986) have sampled this orogen providing a detailed picture of its
lithospheric architecture. In the Iberian Massif, normal incidence (NI) seismic reflection profiles
often acquired with coincident wide angle (WA) reflection/refraction seismic data have
allowed scientists to depict its crustal structure, infer its P and S waves velocity distribution,
place constraints on its geodynamic evolution, visualize the accommodation pattern of





shortening at different crustal levels and, sometimes, deduce the effect of Alpine reactivation
on this Paleozoic orogen.
In this regard, seismic datasets acquired in the Iberian Massif (De Felipe et al., 2020) from the
programs ESCIN (Ayarza et al., 1998, 2004; Pérez-Estaún et al., 1991; Pulgar et al., 1996),
IBERSEIS (Flecha et al., 2009; Palomeras et al., 2009; Simancas et al., 2003), ALCUDIA (e.g.,
Ehsan et al., 2014, 2015; Martínez Poyatos et al., 2012) and CIMDEF (Andrés et al., 2019) have
helped to identify several outstanding features such as, i) clear differences in the intensity and
geometry of reflectivity at upper and lower crustal levels, ii) contrasting deformation patterns,
and iii) a very reflective and sometimes thick lower crust, even in areas where the upper crust
is weakly deformed. In order to explain these features, decoupling of the upper and lower
crust has been invoked along large parts of the sampled area (Simancas et al., 2013). The
existence of a mid-crustal detachment has been addressed as the reason why different
shortening mechanisms exist at different crustal levels. However, coupled crustal deformation
has been inferred for NW Iberia, where large crustal thickening took place. Also, no inference
has been made on how this detachment acted during later Alpine deformation.
In this paper, we present a more complete seismic section of the Iberian Massif. We benefit
from the existence of new datasets (CIMDEF and ALCUDIA WA) that fill the gaps of areas
previously unexplored, like Central Iberia, altogether providing an almost complete transect.
Later on, we revisit the extension and implications of a mid-crustal discontinuity that, in our
view, exists below the entire Iberian Massif, certainly playing a critical role in the decoupling
between upper and lower crustal deformation. Finally, we infer the geometry and nature of
this feature and relate it with the long-debated (e.g., Finlayson et al., 1984; Litak and Brown,
1989; Wever, 1989; Xiaobo and Tae Kyung, 2010) Conrad discontinuity (Conrad, 1925).
**2. Geological setting**
The Iberian Massif represents the westernmost outcrop of the European Variscides, exposing
an almost complete section of this Paleozoic Orogen. It is divided into six zones (Fig. 1; Arenas
et al., 1988; Farias et al., 1987; Julivert et al., 1972) that from N to S and E to W are: Cantabrian
(CZ), West Asturian-Leonese (WALZ), Galicia-Trás-os-Montes (GTMZ), Central Iberian (CIZ),
Ossa-Morena (OMZ) and South Portuguese (SPZ). The CZ and the SPZ represent the external
zones whereas the rest represent the hinterland. The CZ, WALZ and CIZ belong to the northern
margin of Paleozoic Gondwana. The GTMZ represents the remnants of a large nappe stack
formed by pieces of the outermost margin of the Gondwana margin, i.e., a pulled-apart peri-
Gondwanan terrane and oceanic units derived from the oceanic realm separating them. They
were emplaced above the autochthonous CIZ in NW Iberia and are preserved as a large klippen
at the core of late Variscan synforms (Martínez Catalán et al., 1997, 2007; Ries and Shackleton,
1971). Its rootless suture (ophiolitic units) is thought to represent a branch of the Rheic
oceanic realm (Martínez Catalán et al., 1997). The OMZ has been interpreted as a continental
fragment that rifted and probably drifted away from Gondwana (Matte, 2001; Robardet,
2002), docking back later to the CIZ giving rise to a suture and thus representing another peri-
Gondwanan terrane (Azor et al., 1994; Gómez-Pugnaire et al., 2003; Simancas et al., 2001).
Finally, the SPZ is thought to be separated from the OMZ by the Rheic Ocean suture (Simancas
et al., 2003) which was later overprinted by early Carboniferous extension accompanied by





mafic intrusions (Azor et al., 2008) and by transpression (Pérez-Cáceres et al., 2015). In this
context, the basement of the SPZ represents a fragment of Avalonia (Braid et al., 2011, 2012;
Pereira et al., 2014; Rodrigues et al., 2015), and its correlation with the Rheno-Hercynian Zone
in Germany (Franke, 2000; Franke et al., 1990) further supports this affinity.
From a tectonic point of view, the Iberian Massif shows evidences of pre-Variscan activity. The
Cadomian Neoproterozoic event is characterized by continental arc magmatism, deformation
and metamorphism (Bandrés et al., 2004; Dallmeyer and Quesada, 1992; Ochsner, 1993). It
developed above a previous non-outcropping continental crust, and formed the basement
above which the Ediacaran and Paleozoic sedimentation took place, favored by Ediacaran-
Cambrian and Cambro-Ordovician rifting which developed a wide continental platform
(Linnemann et al., 2008; Sánchez-García et al., 2008, 2010). However, most of the tectonic
features observed in the surface are the result of the Devonian and Carboniferous collision
between Gondwana, some peri-Gondwanan terranes and the Avalonian border of Laurentia,
which resulted in the Variscan Orogen (Matte, 2001). The deformation associated to the latter
was diachronous along the Iberian Massif. Next, we describe it, from N to S (in present
coordinates), together with the most important stratigraphic features.
The CZ (Fig. 1) is an external zone located at the core of the Ibero-Armorican Arc (IAA; Dias and
Ribeiro, 1995; Lotze, 1929; Matte and Ribeiro, 1975; Stille, 1924). It is a thin-skinned thrust and
fold belt with a transport direction towards the foreland in the E, and overprinted by oroclinal
folding giving rise to the IAA (Alonso et al., 2009; Pérez-Estaún et al., 1988). Stratigraphically, it
is characterized by a Precambrian sequence, cropping out at its western part and overlain by a
Paleozoic stratigraphic succession that ranges from the Cambrian to a well-developed
Carboniferous: pre-orogenic up to early Carboniferous, and syn-orogenic in the Upper
Carboniferous (Sánchez de Posada et al., 1990; Truyols et al., 1990). Scarce tholeiitic and
alkaline magmatism is related to Cambro-Ordovician rifting (Corretgé and Suárez, 1990), and
no regional metamorphism accompanied deformation, indicating shallow crustal conditions. In
this area, deformation began at ~325-320 Ma, in the Late Mississippian (Dallmeyer et al.,
1997), and the emplacement of nappes that characterize the deformation and the formation
of folds within each sequence took place between the Westphalian B and the Stephanian (313-
300 Ma), in the Upper Carboniferous (Pérez-Estaún et al., 1988). An extensional episode
related to the end of the orogeny led to the formation of Permian Basins (Martínez García,
1981). Later on, extension related to the opening of the Bay of Biscay triggered the
development of deep Cretaceous basins (Quintana et al., 2015; Rat, 1988). Alpine tectonics
uplifted the Pyrenean-Cantabrian range from the end of the Late Cretaceous to the Miocene
(De Felipe et al., 2019; Teixell et al., 2018 and references therein) reactivating Variscan thrusts
and Mesozoic normal faults (Gallastegui et al., 2016).
The WALZ lies to the W of the CZ (Fig. 1). Stratigraphically it consists of a Neoproterozoic
terrigenous sequence uncomformably overlain by a Paleozoic platform succession that ranges
from the Cambrian to the Lower Devonian (Martínez Catalán, 1985), much thicker than that of
the CZ. These sediments were actively deformed along three compressional (C1, C2, and C3)
and two extensional (E1 and E2) phases during the Variscan Orogeny (Martínez Catalán et al.,
2014). Large E vergent folds witness the C1 related compression (360-340 Ma). Those were
later affected by E vergence thrusts resulting from ongoing shortening (345-325 Ma). Crustal



thickening followed by thermal relaxation led to syn-orogenic extension during E1 (330-315
Ma). A last compressional episode (C3, 315-305 Ma) produced upright folds associated with
wrench shear zones while simultaneous extension (E2, 315-300 Ma) continued, characterizing
the latest stages of the orogeny in this area. Crustal melting triggered by compression and
thickening led to extension and to the intrusion of granitoids in the western part of the WALZ.
Thermal models show that the crust could have started to melt within 30 Ma after the start of
crustal thickening, which is then constrained by the ages of Variscan granitoids (Alcock et al.,
134   2009).

The CIZ is the largest of the Iberian Massif zones. Curvature of magnetic anomalies and that of
early (C1) Variscan folds depict the Central Iberian Arc (CIA; Martínez Catalán, 2011a, 2011b),
partly explaining the width of this internal zone. The stratigraphic sequence differs from N to S:
Ordovician felsic metavolcanic, subvolcanic and intrusives (Diez-Montes et al., 2010) represent
the most ancient lithologies cropping out in the N, defining the 'Ollo de Sapo' domain whereas
to the S, Upper-Proterozoic-Lower Cambrian metasediments outcrop (Díez Balda, 1986; Díez
Balda et al., 1995), defining the 'Schist-Greywacke Complex' domain. The pre-orogenic
sedimentary sequence continues to the Devonian, followed by a syn-orogenic Carboniferous
sequence (Martínez Catalán et al., 2004; Robardet, 2002). This area represents a relatively
stable Gondwana margin characterized by the Early Ordovician extension that opened the
Rheic Ocean and allowed intrusion of essentially felsic magmas (Díez Montes et al., 2010). The
deformation phases described for the WALZ affected most of the CIZ although C1, C2 and E1
are somewhat older, according to the propagation of deformation from the hinterland. Slight
differences in the importance of phases can also be found to the center and S (Martínez
Catalán et al., 2019), allowing the CIZ to be divided in two zones. In the NW, intense
recumbent C1 folds and important C2 thrusts exist, related to the emplacement of the GTMZ.
Outcropping rocks show epi- to catazonal metamorphism and ductile detachments. Gneiss
domes of both E1 and E2 extensional phases exist, evidencing significant crustal thinning, and
Variscan granitoids are abundant. To the S, C1 folds are upright, C2 deformation is limited to
the southernmost part, and upright C3 folds are the most important structures (Martínez
Catalán et al., 2012; Martínez Poyatos, 2002). Metamorphism is generally weak and the
amount of granitoids decreases, except in the Iberian Central System (ICS). Here extension
postdates C3 upright folding and thus, it is considered E2.
Alpine tectonics in the CIZ reactivated previous Variscan fractures and triggered the
development of the Iberian Central System (ICS) mountain belt (de Vicente et al., 1996),
allowing the products of syn- and post-E2 crustal melting to outcrop in large areas.
The GTMZ is represented by five klippen that are the remnants of the emplacement of a thick
nappe stack on top of the CIZ. This includes, from bottom to top, a relatively distal part of the
northern Gondwana margin (Parautochthon), the outermost edge of that margin (Lower
Allochthon), a few oceanic units of Cambro-Ordovician and Lower Devonian age (Middle
Allochthon) and a peri-Gondwanan terrane with magmatic evidences of Cambro-Ordovician
rifting and a continental arc setting (Upper Allochthon). Several units show high-P
metamorphism reflecting subduction of the ocean represented by the Middle Allochthon and
involving also the Upper and Lower ones (Arenas et al., 2007; Gómez Barreiro et al., 2007;
Martínez Catalán et al., 2007; Sánchez Martínez et al., 2007). Ongoing subduction during most



of the Devonian (400-365 Ma) built an accretionary wedge that was subsequently emplaced on
top of the CIZ during the early Carboniferous (C2 event, c. 360-340 Ma).
The boundary between the CIZ and the OMZ (the Badajoz Córdoba Shear Zone) has been
largely interpreted as a suture (Gómez-Pugnaire et al., 2003; Simancas et al., 2001), although
no true oceanic units have been identified. It includes amphibolites of oceanic affinity from the
early Paleozoic, as well as eclogite relics. In SW Iberia, outcropping lithologies range from the
Upper Precambrian to the Upper Carboniferous, with an angular unconformity at the Lower
Carboniferous. In the OMZ, the Serie Negra is a thick Neoproterozoic sequence that includes
graphitic quartzites and schists and underwent Cadomian arc-related magmatism and regional
metamorphism (Dallmeyer and Quesada, 1992; Ochsner, 1993; Quesada and Dallmeyer, 1994).
The pre-orogenic Paleozoic sequence is rather complete and was deposited at the peri-
Gondwanan platform, as for the CIZ, although differences in the faunal content and in the
Paleozoic facies, generally more pelitic in the OMZ, point to a more distal position (Robardet,
2002; Robardet and Gutiérrez Marco, 1990). Ediacaran-Cambrian and Cambro-Ordovician
magmatism reflects two rifting events. The latter is the most important one, it includes alkaline
magmatism and is related with the opening of the Rheic Ocean (García Casquero et al., 1985;
Ochsner, 1993; Sánchez-García et al., 2008, 2010). The first deformation event, of Devonian
age, formed overturned and recumbent folds and thrust faults with SW vergence (Expósito et
al., 2002, 2003). Syn-orogenic, early Carboniferous basins developed in an extensional context
and are related to calc-alkaline volcanism and magmatism (Casquet et al., 2001). These
deposits unconformably overlay the early folds and thrusts. Later, deformation continued with
middle and upper Carboniferous sinistral transpression and associated upright NW-SE folds.
A salient seismic reflector, the Iberseis Reflective Body (IRB, Carbonell et al., 2004; Simancas et
al., 2003) seems to be the result of a mantle-derived intrusion located along a mid-crustal
detachment around 350-340 Ma. It was emplaced in the context of early Carboniferous
extension in the SW of the Iberian Massif, while the hinterland to the NW was undergoing the
first stages of compression (C1). Magmatic activity in the SW triggered a high-T/low-P
metamorphism that, otherwise, has a low grade elsewhere in the OMZ (Díaz Azpiroz et al.,
2006; Pereira et al., 2009).
The boundary between the OMZ and the SPZ has been long understood as a suture on the
basis of geometric assumptions (e.g., Carvalho, 1972). Later evidences have reinforced this
point of view suggesting that the above mentioned boundary represents the remnants of the
Rheic Ocean, although Carboniferous transtension and transpression have largely obliterated it
(Pérez-Cáceres et al., 2015 and references therein). The SPZ is a Variscan foredeep basin
strongly deformed by thin-skinned thrust tectonics, and is usually correlated with the
Rhenohercynian Zone of Kossmat (1927) in the Bohemian Massif. It features wide outcrops of
low or very low grade Devonian phyllites, quartzites and sandstones overlain by a lower
Carboniferous (Early Mississippian) volcano sedimentary sequence topped by middle and
upper Carboniferous flysch (Oliveira, 1990). From a tectonic point of view, it is characterized by
Carboniferous S vergent thrusts and folds, the latter featuring axial traces oblique to the
northern boundary of the zone, evidencing transpression (Simancas et al., 2003 and references
therein). Deformation propagated towards the S along the lower and upper Carboniferous
(Oliveira, 1990).



Although the start of the Variscan collision seems to have been frontal or maybe right-lateral
in most of Europe (Shelley and Bossière, 2000), surface geology and interpretation of seismic
data evidences the existence of relevant left lateral transpression and oblique-slip syn-
metamorphic shear zones in the OMZ, SPZ and their boundaries (Pérez-Cáceres et al., 2016;
Simancas et al., 2003 and references therein). In the OMZ, folds and thrusts witnessing
Devonian and early Carboniferous compression are oblique to the OMZ/SPZ boundary,
indicating a transpressional setting. These features are disrupted by later Mississippian
transtensional tectonics (Expósito et al., 2002) that gave way to the intrusion of the Beja-
Acebuches mafic and ultramafic rocks (Azor et al., 2008). Convergence resumed soon after,
leading to the emplacement of the Beja–Acebuches unit onto the OMZ (Pérez-Cáceres et al.,
2015). Inside the OMZ, Devonian and Carboniferous left lateral deformation accounts for ~400
km, higher than perpendicular shortening. Likewise, inside the SPZ, left-lateral displacement is
estimated to reach 90 km whereas the orthogonal one amounts ~60 km (Pérez-Cáceres et al.,
226 2016).

**3. Geophysical setting: Existing datasets, their reprocessing and a brief description**
**3.1. Seismic datasets sampling the Iberian Massif**
Since the early 90's, the Iberian Massif has been sampled by different controlled source
seismic experiments (De Felipe et al., 2020): the ESCIN (1991-1992), IBERSEIS (2000 and 2003),
and ALCUDIA (2007-2012) experiments acquired normal incidence (NI) and coincident wide-
angle (WA) data. The latest project, carried out with the target of understanding the structure
and effect of the Alpine reactivation across the central part of the Iberian Massif, is the
CIMDEF experiment (2017-2019). It acquired densely spaced controlled source WA reflection
and natural source (earthquakes and noise) seismic data. However, the acquisition of NI data
has not currently been planned along this transect, regardless of its potential quality and
relevance, due to the relatively high costs of this kind of experiments.
From N to S, and from E to W, the ESCIN project sampled the northern part of the Iberian
Massif (Fig. 1). Profile ESCIN-1 (1991) is an onshore E-W line crossing the CZ from its eastern,
most external part to its boundary with the WALZ to the W; Profile ESCIN-2 (1991) is an
onshore N-S profile crossing the most external and eastern part of the CZ and reaching the
northern end of the Duero Basin (DB) to the S, which represents the Cantabrian Mountains
foreland basin. The ESCIN-3 (1992) profiles sampled the WALZ and the CIZ along the northern
Iberia shelf. Although it consists of three parts (ESCIN-3.1, 3.2 and 3.3) only the easternmost
ones (3.2 and 3.3.) are relevant for the study of the Variscan crust and thus, included here.
ESCIN-3.3 crossed the entire WALZ to its western boundary with the CIZ, which in this area was
surveyed by the ESCIN-3.2. Geographically, the latter also sampled the allochtonous GTMZ.
But as this is an offshore profile, it shows no evidences of the presence of the GTMZ, and most
of the imaged crust corresponds to that of its relative autochthon, the CIZ.
A significant geographical and methodological gap exists between the ESCIN profiles to the N
and the location of the CIMDEF experiment (Fig. 1). The latter crosses central Iberia from the N
part of the CIZ, then samples the DB down to ICS, and goes on S across the Tajo Basin (TB) till it
reaches again the CIZ metasediments to the S of the ICS.


In the southern part of the Iberian Massif, the onshore ALCUDIA seismic line (NI and WA),
striking NE-SW, was acquired across the CIZ, going from the S of the ICS to the boundary with
the OMZ. Finally, the NE-SW IBERSEIS dataset (NI and WA) is also an onshore profile that
partially overlaps the same structures as the SW end of the ALCUDIA line although with some
50 km of offset to the W. This seismic line samples the southern part of the CIZ, the OMZ and
the SPZ.
Altogether these seismic profiles account for a ~1500 km long seismic transect geared to
understand the crustal and, in places, lithospheric structure of the Iberian Massif and to
constrain its evolution.
**3.2. Processing of datasets**
The data used in this work have been acquired at different times, have different characteristics
(onshore and offshore) and accordingly exhibit very heterogeneous quality. Table 1 shows the
acquisition parameters of all these datasets. The most outstanding differences are: i) the
quality and characteristics of the offshore (ESCIN-3) vs the onshore data, ii) the difference
between the low fold (30) ESCIN-1, ESCIN-2 and ESCIN-3 data acquired with an explosive
source and airguns respectively and the high fold (>60) IBERSEIS and ALCUDIA datasets, which
used Vibroseis trucks as source of energy, and iii) the fact that the CIMDEF dataset lacks NI
data and only provides lower resolution noise and earthquake data, since WA profiles are, as
yet, un-interpreted. Thus, reprocessing the NI data was mandatory, at least at stack and post-
stack level. Figure 2 shows the processing flow followed to homogenize the display of datasets
while preserving the true amplitude (Martínez García, 2019). The software package used for
reprocessing was GLOBE Claritas (www.globeclaritas.com/) and the most important steps were
related with frequency filtering, amplitude weighting and equalization, Kirchhoff time
migration and coherency filtering (Fig. 2). In addition, up to 20 multi-trace attribute analysis
were tested with the goal to enhance structural and lithological impedance contrasts that
allowed to improve the interpretation (Chopra and Alexeev, 2005; Taner and Sheriff, 1977).
Although this methodology has been mostly used in sedimentary reservoirs, we have seen that
the application of these techniques can enhance the continuity of reflections and help to
identify different types of crust, thus easing the interpretation. Some of the results of this
attribute analysis are included in the interpretations.
**3.3. Description of the seismic sections**
The NI datasets included in this paper have already been presented, so the reader will be
referred to previous publications for detailed descriptions of pre-stack processing and
interpretations. Here we will just focus on those features that are essential to our
interpretation.
Reprocessed migrated sections and their interpretations are presented in figure 3 (ESCIN-1),
figure 4 (ESCIN-2), figure 5 (ESCIN-3.3), figure 6 (ESCIN-3.2) figure 7 (ALCUDIA) and figure 8
(IBERSEIS). The description of sections will be done from N to S and from E to W. The CIMDEF
dataset will be described in the discussion (Figs. 9 and 10) as it is key to understanding the
geometry of the mid-crustal discontinuity, its late Variscan reworking and its Alpine
reactivation.



### 3.3.1. Cantabrian Zone (ESCIN-1 section)


The ESCIN-1 section is a ~130 km long, E-W profile crossing the CZ from its most external part
to the Narcea Antiform to the W, in the boundary with the hinterland (WALZ, Figs. 1 and 3). It
consists of two slightly overlapping parts, 1.1 and 1.2, separated a few kilometers in the N-S
direction. The complete ESCIN-1 section migrated at v=5600 m/s (Fig. 2) and its interpretation
are presented in figure 3.
This section was first described and interpreted over an unmigrated image by Pérez-Estaún et
al. (1994). Later works revisited the interpretation, adding travel-time modeling to help on the
understanding of the unmigrated data (Gallastegui et al., 1997). The reader is referred to these
papers for further details than those provided here.
In the upper crust, the western part shows W-dipping reflections that represent the Variscan
imbrication of the basement under the Paleozoic sequence (a in Fig. 3), indicating the
proximity of the hinterland (WALZ). In fact, a Neoproterozoic, non-metamorphic sequence
outcrops in this area, which is probably underlain by an older crystalline basement. One
prominent reflection (a') crosscuts subhorizontal ones, defining a pattern that might indicate
its slightly out of the plane provenance. To the E, the thin skinned tectonics characteristic of
this external zone can be interpreted from shallow subhorizontal to W dipping reflections (b).
The main one among these, running at around 5 s (TWT), is interpreted as the sole thrust of
the thin-skinned orogenic wedge (c). To the W, it gets involved in the crustal ramp observed at
the Narcea Antiform, suggesting that it ends down rooting into the upper part of the lower
crust (d). A low reflectivity wedge of undifferentiated basement (e) located between 4-5 and
8.5 s (TWT) exists underneath the easternmost reflections. This may image some pre-Paleozoic
basement that is interpreted as upper crust, since the pattern of reflections changes below,
suggesting that a significant boundary occurs underneath.
The lower crust shows little reflectivity but seems to be present in the interval between 8.5-14
s (TWT) in the E and between 8.5 and 12 s (TWT) in the W. It features subhorizontal (f) and W
dipping reflectivity to the E, the latter (f') crosscutting the former reflections. These might
represent the imprint of Alpine tectonics over a previously deformed/reflective lower crust. To
the W, reflectivity seems to be subhorizontal or dipping to the E (g). Some of the dipping
reflectivity observed at the edges of section ESCIN-1 might be related to the migration effects
over a little reflective section and caution should be taken when interpreting it.
The Moho along this section is located at nearly 14.5 s TWT (~45 km) in the eastern part, and
shallower (12 s TWT, ~36 km) to the W. The crustal thickening observed to the E (h) is probably
related with an out of section image of the crustal Alpine root, better observed in profile
ESCIN-2, which is described next.

### 3.3.2. Cantabrian Zone and Duero Basin (ESCIN-2 section)


The ESCIN-2 seismic line is a 65 km long, N-S section that samples the transition between the
CZ and the DB (Fig. 1). Even though this profile was geared to study the Alpine structures, it
shows how the Variscan features have been inherited and reactivated during the Cenozoic
compression between the Iberian Peninsula and the European plate. The section was first





presented by Pulgar et al. (1996). Later on, some authors have used this image to constraint
the Alpine structure in the North Iberian Margin (e.g., Fernández-Viejo et al., 2000; Gallastegui
et al., 2016). However, only Teixell et al. (2018) used a migrated version (4000 m/s) of this
section. Here we present the results of a Kirchhoff time migration at 5600 m/s (Fig. 4).
This seismic line shows, in places, a conspicuous reflectivity that allows a straightforward
interpretation. To the S end, the upper crust is characterized by high amplitude horizontal
reflectivity representing the DB sedimentary sequence. It occupies the interval from 0-3.5 s
TWT (a in Fig. 4) and appears to be offset by N dipping reflections (b). The latter have been
interpreted as S vergent Alpine thrusts affecting the CZ basement and partly the DB sediments.
The rest of the crust is less reflective although N dipping reflectivity (c), also interpreted as
imaging Alpine thrusts, crosscuts shallow subhorizontal weak reflections that represent the
Paleozoic sedimentary sequence of the CZ (d).
The lower crust presents higher amplitude reflectivity. In general, a thick band of horizontal
reflections located between 7.5 and 12 s (TWT) at the southern part of the profile, bends and
dips to the N in the northern part of the line (e) in response to Alpine compression. Although
the stacked section shows that this N dipping reflectivity reaches 14.5 s TWT (Pulgar et al.,
1996) the migrated sections (Teixell et al., 2018 and Fig. 4) indicate that these reflections move
southward and upward to less than 14 s (TWT), while losing amplitude and coherence. In fact,
the geometry of the bottom of the lowermost crust (Moho) is deduced on the basis of the
geometry of its uppermost part, the lower crust internal reflectivity, and the amplitude
contrasts observed in the attribute analysis (Fig. 4). Furthermore, its depth is solely stablished
on the basis of the position of the strongest subhorizontal reflections to the S.
Even though this profile shows the imprint of recent Alpine shortening, no reflections are
observed to crosscut the entire crust. In contrast, reflectivity suggests that deformation is
decoupled between the upper and lower crust. However, this section is not long and/or
reflective enough as to image where the Alpine thrusts (c) root. Possibly, they merge into the
roof of the underthrusted CZ lower crust. In addition, the migration effects on the edges of the
section produce misleading reflectivity than hinders more detailed interpretations.
**3.3.3. West Asturian-Leonese Zone (ESCIN-3.3 section)**
The ESCIN-3.3 profile is part of a ~375 km long crooked offshore seismic line consisting in
ESCIN-3.1, 3.2 and 3.3. The latter is 137 km long, parallel to the coast and close to it across the
WALZ (Fig. 1). It was first presented by Martínez Catalán et al. (1995) and Ayarza et al. (1998,
2004). Later on, its image has been used to constrain the structure of the western North
Iberian Margin and that of the transition between the WALZ and the CIZ (Martínez Catalán et
al., 2012, 2014).
Reflectivity in the upper crust is characterized by the image of Mesozoic sedimentary basins (a
in Fig. 5) related to the extension that led to the opening of the Bay of Biscay. Underneath, W
dipping reflections are interpreted as the imprint of the first stages of Variscan compressional
deformation in the WALZ (C1 and C2), developing E-vergent thrust faults (b). These affect the
pre-Paleozoic basement and root in the upper part of a thick reflective band interpreted as the
lower crust (c) or in a sole thrust (d) that also reaches the lower crust.



The lower crust (c) is represented by a thick band of subhorizontal reflectivity (8-12 s TWT)
that thickens (6-12 s TWT) in the westernmost part of the WALZ (CDP 3000) underneath the
Lugo Dome, an extensional structure bounded to the E by the Viveiro normal fault (Fig. 5).
Then it thins towards the end of the line, when entering the CIZ, thus defining a Moho offset of
~2 s TWT. The ESCIN-3.3 lower crust seems to feature an internal layer with mantle P-wave
velocities when modeled from coincident WA data. Accordingly it was interpreted as consisting
of the WALZ lower crust underthrusted by the CZ lower crust (Ayarza et al., 1998). This model
would compensate the high shortening observed in the upper crust of the CZ, a thin-skinned
belt whose sole thrust roots at the contact with the WALZ. However, the ESCIN-3.3 WA data
need to be revisited as it is a fan profile difficult to model with old conventional 2D algorithms.
The internal reflectivity of the lower crust shows W dipping reflectors (e), similar to the ones
observed in the upper crust and probably imaging Variscan deformation in the lower crust,
either compressional or extensional. They crosscut subhorizontal reflectivity, thus postdating
it.
Even though migration over discontinuous reflections blurs the seismic section in the edges,
reflectivity never seems to cross-cut the crust and/or the Moho, indicating that deformation is
decoupled at upper and lower crustal level. Subcrustal E dipping reflections are interpreted as
the out-of-the-plane image of the Alpine southward subduction of the Bay of Biscay oceanic
crust (Ayarza et al., 1998, 2004), which is out of the scope of this paper.
The boundary between the WALZ and the CIZ is the Viveiro Fault, one of the most striking
surface expressions of Late Variscan extensional tectonics. To its W, gravitational collapse of a
thickened crust and associated crustal extension and melting have played a key role in the
orogenic evolution of the CIZ. However, to the E, crustal re-equilibration after C1 and C2
thickening was less important and igneous activity decreases. Even though this fault itself is
not identified in the seismic section (Fig. 4), the reflectivity in general varies on both sides of it,
featuring a thinner (9 s TWT vs 12 s TWT) and more transparent crust to the W. In fact the
geometry of some reflections (f) in the boundary between the WALZ and the CIZ, above the
thickest lower crust, and the subtractive way the sole thrust (d) merges with the lower crust
(d') seem to indicate the effect of extensional tectonics, sometimes reactivating compressional
structures (d-d'). Such a reactivation has been described for the base of the main thrust sheet
in the WALZ based on structural and metamorphic considerations (Alcock et al., 2009).
Conversely, further to the E of the section, reflectivity probably represents the geometry of
preserved compressional deformation.
**3.3.4. Northern Central Iberian Zone (ESCIN-3.2 section)**
The seismic line ESCIN-3.2 is a 97 km long profile, also parallel and close to the coast, and
sampling the relative autochthon to the GTMZ, i.e. the CIZ (Figs. 1 and 6). It was first described
by Álvarez-Marrón et al. (1996) and later by Ayarza et al. (2004).
This profile shows, in the upper part, a band of high subhorizontal reflectivity related to
Mesozoic basins, as in profile ESCIN-3.3 (a in Fig. 6). The rest of the upper crust is not very
reflective although a couple of W-dipping reflections (b) rooting in a thin band of strong
reflectivity are observed. These reflections, located in the E of the section from 4.5 s to 8 s
TWT, define a sort of duplex, extensional or compressional, but later extended, indicating in



any case boudinage and crustal thinning. To the W, the upper crust is very transparent and just
a few weak reflections can be observed.
The narrow reflective band at 8-9 s TWT represents the lower crust, and is the most striking
feature of this profile. This 1 s TWT thick feature contrasts with the neighboring ESCIN-3.3 and
even ESCIN-1 sections, which show a much thicker lower crust (4-5 s TWT). Reflectivity in this
band is subhorizontal (c), although somewhat undulated, while the band itself is slightly
inclined to the W. In the E, the Moho is located at around 9 s TWT (~27 km), the shallowest
identified so far in the Iberian Massif. Subcrustal dipping reflections (d) are again associated to
the 3D image of the southward subduction of the oceanic crust of the Bay of Biscay during the
Alpine convergence. They have been already modeled by Ayarza et al. (2004) and will not be
further discussed in this paper.
This profile samples the northern CIZ, where Variscan crustal thickening during C1 and C2, was
most important. Consequently, later gravitational collapse triggered extensional tectonics and
crustal melting, allowing the intrusion of granites and the development of extensional
detachments (with associated metamorphic offsets). The image of line ESCIN-3.2 shows a
transparent upper crust to the W suggesting that granites occupy most of it, which is
supported by onland geological mapping. Some thrust faults, as those imaged by W-dipping
reflections in profile ESCIN-3.3, probably root here and are represented by the W-dipping
reflections (b) at the base of the upper crust. However, these were later flattened and/or
reactivated as extensional detachments by crustal thinning. The narrowness of the highly
reflective lower crust here suggests that crustal thinning was largely accommodated at this
level, as the upper crust has basically the same thickness as in the ESCIN-3.3 line (up to 6-7 s
TWT). In addition, crustal melting might have also affected the top of the lower crust. But even
though large parts of the crust were melted, reflectivity exists deep in the upper crust,
suggesting that crustal melting was nor pervasive and/or reflectivity is linked to syn- or late-
tectonic features.

### 3.3.5. Southern Central Iberian Zone (ALCUDIA section)

The ALCUDIA seismic profile was first presented by Martínez Poyatos et al. (2012) and
processed and further interpreted by Ehsan et al. (2014). It is a more than 220 km long, NE-SW
seismic profile sampling the CIZ to the S of the ICS down to the boundary with the OMZ (Fig. 1).
This profile presents a fairly transparent upper crust where scarce reflectivity to the S is related
to the boundary (suture) between the CIZ and the OMZ, namely, the Central Unit (CU; a in Fig.
7) and to the presence of vertical folds (b). Some very transparent zones (c) appear to be in
relation with the existence of granitic batholiths. To the N, the intrusion of these granites,
associated to the existence of normal faults (d), is one of the evidences of extensional
tectonics affecting the southern part of the CIZ. The rest of the upper crust shows weak and
discontinuous reflectivity that responds to the existence of vertical folding affecting lithologies
with little impedance contrast.  In fact, deformation in the upper crust of this part of the CIZ is
weak, with absence of low-dipping structures typical of tangential tectonics.
The lower crust shows a very different image to that of the upper crust. It is a thick band, of up
to 6 s TWT (from 4 s to 10 s), of mostly subhorizontal high amplitude reflectivity (e) that at



some points appears to be cut across by N-dipping reflectors (f). S-dipping internal reflectivity
is also identified although more scarce (g). The lower crust thins in the northern end of the
profile, near the ICS, where intrusion of granites and other evidences of crustal re-equilibration
suggest that extension played a key role. Accordingly, we suggest that the mechanisms that
triggered this lower crustal thinning are related with melting and extension and not with
compression, as previously proposed (Ehsan et al., 2014; Martínez Poyatos et al., 2012), and
that the N dipping reflectivity observed above the lower crust in that area (h) is the expression
of extensional tectonics.
One of the most striking features of this profile is the crocodile-like structure affecting the
lower crust at around CMP 10000 (f). This structure, most likely related to Variscan shortening,
accommodates an important part of the deformation at lower crustal level and evidences that
sub-horizontal reflectivity of the lower crust is pre-Variscan, thus raising the question about its
precise age and origin. Despite the presence of this feature in the depth continuation of the
suture between the CIZ and the OMZ, reflectivity does not crosscut the whole crust, suggesting
the existence of a detachment in the top of the lower crust. This contrasts with the presence in
the upper crust (CU) of retro-eclogites with peak metamorphic conditions of 19 kbar and
~550°C (López Sánchez-Vizcaíno et al., 2003). Finally, the Moho boundary is located at a fairly
constant depth (~10 s TWT, i.e. 30-33 km), although the lower crust seems to be preserved and
a local crustal imbrication into the mantle is observed underneath the crocodile-like structure.

### 3.3.6. Central Iberia, Ossa-Morena and South Portuguese Zones (IBERSEIS section)

The IBERSEIS seismic line was first presented by Simancas et al. (2003). A number of later
works added information and details to its interpretation (e.g., Carbonell et al., 2004;
Schmelzbach et al., 2007, 2008; Simancas et al., 2006). This section crosses the southernmost
CIZ, the whole OMZ, and most of the external SPZ (Fig. 1). It samples two major boundaries
interpreted as suture zones: that between the CIZ and the OMZ (CU, Azor et al., 1994) and the
one bounding the OMZ and the SPZ, which has been largely affected by young Carboniferous
events (Pérez-Cáceres et al., 2015). The IBERSEIS profile structurally overlaps the ALCUDIA
profile along ~30 km, but it is displaced some 50 km to the W. Its interpretation is shown in
figure 8.
This section is ~300 km long and features an outstanding reflectivity at upper and lower crustal
levels. In the upper crust, a wealth of N dipping reflections (a in Fig. 8) image a S verging thrust
and fold belt. In the SPZ, the most reflective events are probably related with normal faults
derived from the extension that led to the opening of the Rheic Ocean and were later
reactivated as thrusts during the Late Carboniferous compression. Some authors link the
highest reflective features to the middle Carboniferous volcano-sedimentary complex
(Schmelzbach et al., 2008), which might have used these fractures as a conduit, thus enhancing
reflectivity. In the OMZ, N dipping reflections probably image Variscan thrust faults as some
coincide with such mapped structures. Their lesser reflectivity might indicate the lack of
involvement in the thrusts of lithologies that increase the impedance contrast.
Upper crustal reflectivity in both, the ZOM and SPZ, does not cross to the lower crust, rooting
at a mid-crustal level that, in the SPZ is transparent and does not have any particular
expression itself. However, in the OMZ, a reflective layer exists at this depth (b): it has been





defined as the IBERSEIS reflective body (IRB, Simancas et al., 2003), a 140 km long, high
velocity conductive feature (Palomeras et al., 2009) that is supposed to represent an early
Carboniferous mantle-derived intrusion. Its origin has been related to mantle plume activity
that thinned the lithosphere and extracted mantle-derived melts from the ascending
astenosphere (Carbonell et al., 2004). Its surface expression are intraorogenic transtensional
features (Rubio Pascual et al., 2013; Simancas et al., 2006). Alternatively, Pin et al. (2008) have
suggested, based on geochemical constraints, a tectonic scenario of slab break-off for this
feature. Internal reflectivity along the IRB is mostly subhorizontal, probably due to the effect of
the intrusion along a subhorizontal detachment, and evidences little imprint from Variscan
deformation. The body is slightly inclined to the S, at odds with the detachment being the sole
thrust of the OMZ upper crustal imbricates. Perhaps it was, but its inclination changed during
subsequent deformation, as later suggested.
The lower crust shows slightly different patterns in the CIZ and OMZ on one side and the SPZ
on the other. In the southernmost part of the CIZ and northern OMZ, N and S dipping
reflections define a wedge (c) that might be the western continuation of the crocodile-like
structure observed in the ALCUDIA seismic line in an equivalent structural position (f and g in
Fig. 7). In this section, the limited crustal imbrication into the mantle identified in the ALCUDIA
line is not observed, perhaps because it only occurs further to the N or E. This structure may be
the reason why the IRB is shallower at this point, indicating that the latter is older than the
crocodile compressional feature. The rest of the lower crust shows S dipping (d) and sub-
horizontal (e) reflectivity that does not exhibit clear crosscutting relationships, thus hindering
their interpretation. However, near the boundary with the SPZ, this reflectivity seems to be
affected by N dipping features (f) overprinting them. In the SPZ, the lower crust shows a more
homogeneous image, with subhorizontal reflectivity (g) that is often cut across by longer scale
S dipping features (h) that postdate them. The latter probably represent fractures that firstly
accommodated the extension linked to the opening of the Rheic Ocean and were then
reactivated as thrusts during the late Carboniferous compression and collision of the SPZ
basement with the OMZ. The most conspicuous of these reflections (h') cuts the IRB in its
southern part and seems to offset the lower crustal upper boundary between the SPZ and the
OMZ.
Even though the lower crust in the OMZ and SPZ shows dipping features, none of them cross
to the upper crust, thus rooting at a mid-crustal level as does the upper crustal reflectivity. This
implies again the existence of a discontinuity in the mid-crust.
Despite of crossing two suture zones and imaging part of a crocodile-like structure, the
IBERSEIS profile shows a fairly flat Moho located at ~10 s TWT, the same apparent depth as in
the ALCUDIA line (30-33 km). Its signature is very clear underneath the SPZ and a bit blurry
below the IRB.
**4. Discussion**
Simancas et al. (2013) already undertook an integrated interpretation of most of the seismic
sections presented here focusing on, i) the accommodation of orogenic shortening at crustal
scale, (ii) the relationships between convergence, crustal thickening and collisional granitic
magmatism, and (iii) the development of the Iberian Variscan oroclines. In this paper the same





sections are presented, but they have been reprocessed at stack level and time migrated using
a Kirchhoff algorithm. In addition, two extra sections that image the alleged mid-crustal
discontinuity after the Alpine reactivation are taken into account. The first one is the N-S
ESCIN-2 NI dataset (Fig. 4), in the CZ, where this discontinuity has remained untouched during
late Variscan evolution but was reactivated during the Alpine Orogeny. The second one results
from the CIMDEF experiment, carried out in the CIZ across the ICS, where the mid-crustal
discontinuity has probably been affected by crustal melting during the Late-Variscan extension
and by later Alpine reactivation. The latter sections somehow fill the gap existing in Simancas
et al. (2013).
Figures 3 to 8 represent an effort to show a homogeneous seismic image of the Iberian Massif
crust that eased its integrated interpretation. Next, we discuss the main observed features,
their implications and how they contribute to the understanding of the structure and evolution
of the Iberian Massif, adding constraints to the origin of the elevation of the central Iberian
Peninsula. Figure 9 presents a simplified sketch of the crustal layers observed in the Iberian
Massif. Figure 10 shows a compendium of the position of the mid-crustal discontinuity and the
Moho depth (in TWT) along the entire Iberian Massif as deduced from seismic NI data together
with a map of the entire Iberian Peninsula Moho depth (Palomeras et al., 2017) that includes
the position of the seismic profiles for comparison. We will refer to these figures along most of
the discussion.
A particular feature of the SW Iberian Massif is the great importance of out-of-section, mainly
left-lateral shear zones associated to its suture boundaries. They displaced central and
northern Iberia to the NW with respect to southern Iberia (Simancas et al., 2013). The seismic
sections do not provide constraints about this movement, as it is perpendicular to their layout.
Thus, interpretations in these areas must be taken with caution.

### 4.1. The upper crust in the Iberian Massif: a depth image of outcropping geology

Most of the seismic sections display a moderate to thick upper crust (4 to 8 s TWT, Fig. 9), with
very variable reflectivity. Reflections have been confidently related to outcropping Variscan
structures. As such, N dipping reflectivity in the SPZ and the OMZ is related to S vergent folds
and thrust faults mapped in the surface. W dipping reflections in the CZ are related to mapped
thin-skinned thrusts. The same type of reflectivity observed in the WALZ, reaching deeper
levels in the crust and rooting in the lower crust, has been addressed as evidence of thick-
skinned thrust tectonics, which in the hinterland affects the pre-Paleozoic basement.
Particularly interesting is the upper crustal SPZ seismic image in contrast with that of the CZ,
both representing external zones. While in the latter thrusts are observed to root in a shallow
sole detachment, in the former one reflections/thrusts root in the lower crust. This feature will
be discussed in the next section.
Only a few seismic profiles feature a transparent upper crust. Lack of reflectivity has been
related to low fold data (ESCIN-1 and ESCIN-2, Figs 3 and 4), and most importantly to the
existence of a re-equilibrated upper crust having recorded large amounts of partial melting, as
shown by voluminous outcropping granitoids (ESCIN-3.2 and N of ALCUDIA, Figs. 6 and 7). The
existence of vertical folds affecting little reflective monotonous lithologies also results in a
fairly transparent upper crust in most of the ALCUDIA section (Fig. 7).



None of the upper crustal reflections observed and interpreted in the presented Iberian Massif NI seismic sections seems to cut across the whole crust, always rooting in a sole thrust (ESCIN-1, Fig. 3) or in the lower crust (the rest of them).

**4.2. The lower crust in the Iberian Massif: accommodation of shortening, extension and its nature**

The Iberian Massif dataset presented here shows a very coherent image of the lower crust. Its reflectivity is high and usually subhorizontal. However, cross cutting relationships with later features of opposite dips evidence a multi-phase origin for this reflectivity.

The SPZ, OMZ, WALZ, CZ and the southern CIZ show that this part of the crust is also thick (4 to 6 s TWT). However, in NW Iberia and the northern part of the ALCUDIA section (Figs. 1 and 7), the few existing NI profiles indicate that the in the northern CIZ, the lower crust is much thinner (1 to 2 s TWT) and irregular (ESCIN-3.2, Figs. 6 and 9). This thin lower crust has been observed in the area characterized by outcropping syn-collisional granitoids (zone II of Simancas et al., 2013). These witness the onset of crustal re-equilibration processes triggered by gravitational collapse, extension and crustal melting during the Late Carboniferous. The straightforward conclusion is to attribute the architecture of this lower crust to late Variscan orogenic extension and melting, implying that crustal thinning has been mostly accommodated by its lowermost part.

Nevertheless, a gap of crustal-scale NI data exists in most of the northern CIZ. The CIMDEF noise autocorrelation profiles (Figs. 1 and 9 and ) show a thick (~5 s TWT) lower crust in most of this area, which essentially corresponds to the CIZ (Andrés et al., 2019 and this volume). This is in conflict with the NI sections ESCIN-3.2 and northernmost ALCUDIA, where the highly reflective lower crust is less than half as thick. However, granitoids are probably scarce in the Variscan basement hidden under the DB, which can then present a thick lower crust. But in and near the ICS, a rather continuous internal reflection in the lower crust could be interpreted as its top part (Figs. 9 and 10), thus indicating that crustal thinning and melting, observed in the surface, has also affected the lower crust (Andres et al., this volume).

Extension in the northern CIZ occurred simultaneously with shortening in the SW Iberian Massif. According to Simancas et al. (2013) this suggests that the tectonic stresses would be dominantly compressional, still induced by ongoing collision. In fact, gravitational instabilities in a thickened crust should mostly be affecting the upper crust. In this context, theoretical models (Royden, 1996; Seyferth and Henk, 2004) indicate that beneath the areas of extension in the upper crust, shortening may prevail in the lower crust. This mechanism is an efficient way for syn-convergent exhumation of deep rocks.

Indeed, from a regional tectonic perspective, compression was active till the end of the Variscan orogeny, and at times, clearly simultaneous with extension (C3 and E2 overlapped in the interval 315-305 Ma; Martínez Catalán et al., 2014). But it is clear that extension affected the lower crust, as it appears thinned in areas of transparent, extended molten crust (ESCIN-3.2 and ALCUDIA sections, Figs. 6 and 7). However, the irregular pattern observed in the ESCIN-3.2 lower crust might indicate the existence of folds in this re-equilibrated layer, witnessing the simultaneity of extension and compression even at lower crustal level (Fig. 6). In addition, we



cannot rule out that these undulations represent boudinage (i.e. extension) or Alpine folding,
although we consider the latter less likely.
In the ALCUDIA section, the imaged part of the CIZ underwent only moderate upper crustal
shortening (Martínez Poyatos et al., 2012). According to Simancas et al. (2013), the thick
laminated lower crust, representing pre-Pennsylvanian (most probably pre-Variscan) ductile
deformation, appears deformed in two sectors near both ends of the profile, concentrating
shortening in discrete structures that compensate the upper crustal deformation. The first of
them is the very conspicuous crocodile-like structure observed in the southern end, and also
imaged in the northern part of the IBERSEIS line (Fig. 9b). This structure mimics localized
crustal indentation of the OMZ into the CIZ, producing a local underthrusting of the latter to
the S that is still (partly?) preserved. Indentation generated tectonic inversion of the Los
Pedroches early Carboniferous basin (Simancas et al., 2013) and bending of the overlying
upper crust, as seen by the uplift of the IRB, both of which predate the imbrication. The Los
Pedroches batholith intruded above at 314-304 Ma in an extensional setting (Carracedo et al.,
2009), postdating the age of the wedge as no further deformation affected the batholith.
Indeed, the crocodile-like feature must represent early Carboniferous Variscan compressional
deformation and must account for part of the shortening observed at upper crustal level.
However, to the NE of this section, a ramp-and-flat geometry has been interpreted as a major
lower crustal thrust (Martínez Poyatos et al., 2012; Simancas et al., 2013) that helps to
compensate upper and lower crustal shortening. However, the highly reflective lower crust is
not repeated in the hanging wall to the structure, so that a subtractive character is a
reasonable alternative. As stated above, the thin lower crust to the N of the ramp seems to be
clear evidence of lower crustal thinning (Fig. 9b), supported by the fact that it underlies an
area of upper crustal extension, the Toledo gneiss dome, characterized by normal faulting and
pervasive partial melting (Barbero, 1995; Hernández Enrile, 1991). Regardless of how much
shortening that area accommodated during crustal thickening and even though the observed
ramp could be a former thrust fault reactivated during later extension, the present image of
the lower crust does not suggests compensation of upper crustal shortening. In fact, the lower
crust in the ALCUDIA section is anomalously thick elsewhere (up to 6 s TWT, 18 km) suggesting
the possibility of ductile thickening previous to the extension that triggered thinning at its
northern part.
In the IBERSEIS profile, lower crustal dominant reflectivity is also subhorizontal but disrupted
by N and S dipping features (Fig. 9b). Whereas in the OMZ these features usually dip to the N,
as do the upper crustal reflections representing Variscan thrusts, in the SPZ they surprisingly
mirror the upper crustal Variscan thrusts, dipping to the S. Furthermore, one of these features,
placed close to the boundary with the OMZ, affects almost the entire lower crust.
Orogenic orthogonal shortening in the OMZ upper crust has been estimated in 120 km (~57%)
and in the SPZ around 80 km (~45%) or even less (Pérez-Cáceres et al., 2016). According to
Simancas et al. (2013), the crocodile structure and a not observed associated northward
subduction of the OMZ might account for this shortening in the OMZ. Similarly, in the SPZ, the
lower crustal imbricated structures represent only ~ 20 km of shortening so that according to
these authors, detached lower crustal subduction along the OMZ/SPZ might have
accommodated the other 60 km.
In this regard, we suggest that the present day SPZ crustal image represents its decoupled
early Carboniferous extension and later compression. This evolution would have erased any
evidences of previous (pre-Carboniferous) subduction, and forced the SPZ to thin during
extension. i.e., the lower crust had to decrease its thickness ductilely, perhaps first in a more
or less distributed way and later through localized shear zones (brittle or not depending on the
depth) as it became shallower. However, the upper crust could have preserved most of its
original thickness, as the developing basins associated to extension would have been
constantly fed by sediments and igneous extrusions and intrusions (like the IRB in the OMZ).
Later compression would have folded and thrusted the upper crust, and also thickened the
lower crust. A few lower crustal normal shear zones might have developed during extension
and then be reactivated as ductile thrusts during compression. Those are today observed as S
dipping reflections that disrupt the subhorizontal previous reflectivity in the lower crust and
mirror thrusts in the upper crust. Accordingly, distributed ductile deformation and thrusting
might have thickened the lower crust back to its original (or simply stable) thickness in the SPZ
and elsewhere, something that cannot be measured but would need to be accounted for when
comparing shortening at upper and lower crustal level. The resulting seismic image of the SPZ
would then be that of an extended and then inverted margin, with mirroring reflectors in the
upper and lower crust merging in a mid-crustal discontinuity and providing a seismic image
different to that of a typical foreland thrust and fold belt (e.g., CZ; Fig. 3). This evolution differs
from that of a hyperextended magma-rich margin as stretching of the upper and lower crust is
not coupled and faults do not cut across the crust and penetrate down into the mantle. In any
case, the S dipping lower crustal reflections, active during the Late Carboniferous, postdate the
sub horizontal reflectivity of the lower crust. It is worth mentioning here that the SPZ seismic
image is identical to that of the Rhenohercynian Massif in Germany (Franke et al., 1990;
Oncken, 1998) suggesting a similar evolution.
The discussion above shows that the lower crust in the Iberian Massif is thick, except when it is
affected by late orogenic extension. The mechanisms that produced lower crustal thickening
are probably related with compressional deformation, mostly ductile. Continental
underthrusting of the CZ underneath the WALZ (Ayarza et al., 1998, 2004), indentation of the
OMZ in the CIZ (Fig. 9b) and Variscan thrust-like structures probably played an important role.
In addition, the latter help to constrain the age of the subhorizontal reflectivity. Frequent
disruption of subhorizontal lower crustal lamination by Variscan (late Carboniferous) dipping
features indicates that the lamination developed prior to Variscan compressional deformation.
What this lamination represents is still an open question.
Many vertical incidence seismic reflection profiles worldwide have shown reflective lower
crusts (e.g., Meissner et al., 2006; Wever, 1989). Lower crust seismic lamination has been
often related to late orogenic extensional events (Meissner, 1989). In the Iberian Massif,
surface geology shows that late orogenic extension affects the upper crust, mainly in areas of
large previous thickening. In contrast to this author's models, important thinning of the lower
crust takes place in those areas (ESCIN-3.2 and northern ALCUDIA, Figs. 6, 7 and 9). Certainly,
lower crustal lamination might come from underplatting eased by extension in magma rich
margins (Klemperer et al., 1986). But also, ductile deformation is a very likely source of lower
crustal lamination. Dipping events observed in the lower crust crosscutting a strong banded
reflectivity represent the latest orogeny-related shortening, which will be further flattened and
horizontalized in the next orogeny. Continuous superposition of deformational events at lower
crustal level managed to decrease the dip of structural/lithological markers and define a
subhorizontal fabric. These deformation mechanisms can generate structures with a strongly
defined anisotropy, which result in a strongly laminated lower crustal fabric (Carbonell and
Smithson, 1991; Okaya et al., 2004). Accordingly, a laminated lower crust may represent an
overly reworked lower crust that has been ductilely deformed over several orogenies.
Opposite to the model by Meissner (1989), such a horizontal reflectivity is observed along the
Iberian Massif in areas where late orogenic extension is absent or weak: the SPZ (Avalonia),
the OMZ (peri-Gondwana), the not extended CIZ, the WALZ and the CZ (Gondwana). Thus, we
suggest that strong lamination in the deep crust is probably a global characteristic of reworked
lower crusts not affected by late orogenic extension in the latest orogeny.
**4.3. The Moho and crustal thickness in the Iberian Massif**
The crust-mantle boundary, i.e., the Moho, is basically flat in the Iberian Massif except where
affected by the Alpine tectonics (Fig. 10). This is rather surprising as the lower crust seems to
be quite well preserved, suggesting that the Moho geometry has been flattened out through
slow, not invasive, readjustments.
Flat Mohos imply the existence of either isostatic and/or thermal, late to post-orogenic
processes that have managed to eliminate crustal roots (Cook, 2002). NW Iberia was affected
by late Carboniferous extension that heated and reworked the CIZ, possibly without significant
mantle involvement (Alcock et al., 2009, 2011), but producing crustal thinning (Palomeras et
al., 2017: see Moho depth map in Fig. 10). Thick and thermally mature crusts might experience
lateral flow of its low-viscosity deeper part that contributed to reduce crustal roots (Seyferth
and Henk, 2004). This process might have partly occurred in the CIZ sampled by the ESCIN-3.2
section (Fig. 6) where an outstanding change in lower crustal thickness and signature exist,
manifested by a thinner and very reflective lower crust in contrast to that to the E, in the WALZ
(1 vs 3 s TWT). In the latter, the Variscan crust is still thick even though it experienced late-
Variscan extension in its western part and the whole area was slightly affected offshore by the
extension linked to the opening of the Bay of Biscay.
In the SW Iberian Massif, a thick laminated lower crust is still observable while the Moho depth
is fairly constant (~10s TWT). Carboniferous-to-Permian isostatic rebound in response to
tectonic thickening, erosion and localized Permian thermal readjustments must have
contributed to flatten the Moho. However, seismic reflections show that crustal imbrication
into the mantle has locally survived post-orogenic Moho resetting. This indicates that isostatic
equilibrium has been reached in a long wavelength scale, but that local features can still
remain if they can be supported by the crustal strength and do not pose an isostatic constraint.
**4.4. The (missing) middle crust in the Iberian Massif (and elsewhere?)**
One of the highlights of this work is the lack of a seismic layer that can be identified with the
middle crust. But, what is the middle crust?



From a metamorphic point of view, the middle crust could be ascribed to the mesozone, which
may be correlated with the amphibolite facies, whose temperature ranges between 400-500
and 600-800°C, the precise limits depending on the pressure (Spear, 1993). In addition, the
epizone, between 200-250 and 400-500°C and typically represented by the greenschist facies,
is also a metamorphic entity which develops during metamorphism under several kilometers
of anchi- and no metamorphic rocks. The depths corresponding to these temperature intervals
vary with the geothermal gradient. For a Barrovian gradient, typical of a continental crust
undergoing collision, the depths for epizone and mesozone can be estimated around 10-20
and 20-30 (± 5) km respectively. However, the boundaries of these metamorphic zones might
have a gravity, i.e. density signature, but not a seismic one. Furthermore, epi-, meso- and
catazonal rocks outcrop everywhere in any eroded orogenic belt, implying that they do not
represent a seismic mid crust but actually occur in the upper crust.
From a seismic point of view, a middle crust could be a crustal level bounded in its upper and
lower parts by characteristic reflections that indicate the existence of important impedance
contrasts at its top and bottom. In this regard, only the IRB, intruded between the upper and
the lower crust (Carbonell et al., 2004; Simancas et al., 2003), and providing conspicuous
velocity contrasts (Palomeras et al., 2009, 2011) fulfill that requirement. However, it is most
probably an intrusion emplaced at a mid-crustal discontinuity and does not represent the
middle crust.
WA reflection seismic data from the northern Iberian Massif have often resulted in
multilayered models despite weak evidences of continuous reflectivity at these levels (Ayarza
et al., 1998; Fernández-Viejo et al., 1998, 2000; Pedreira et al., 2003). Even though local
velocity contrasts capable of providing weak and patchy reflectivity contrasts exist at different
crustal depths (e.g., thrust faults and normal detachments may represent lithological
boundaries with a noticeable velocity contrast), these are not orogen-scale features but local
reflectors. Many of these reflections observed, interpreted and/or extrapolated as middle
crust in seismic WA datasets belong in fact to the upper crust, when compared with NI data,
i.e., lie above the mid-crustal discontinuity. In this regard, the short wavelength
heterogeneities of the crust can be seen by low resolution WA dataset as laterally continuous
features (Levander and Holliger, 1992), something that has led us to wrong models.
According to the above we argue that, in the Iberian Massif, no seismic middle crust can be
identified. In the hinterland, reflectors imaging deformation in the upper crust root in the top
of the lower crust. Only in ESCIN-1, which depicts the thin-skinned deformation of the CZ,
thrust faults root in a sole thrust and one could argue that the basement underneath these
shallow reflections represents the middle crust. But in the shallower part, to the E, early
Paleozoic and Neoproterozoic sediments occur on both sides of the sole thrust. Also, in the
deeper parts, to the W, the previous crystalline basement is probably involved in imbrications
affecting the upper crust. Thus, in our opinion, that non-reflective basement represents the
upper crust.
In the Iberian Massif, the Paleozoic was deposited unconformably above Neoproterozoic
sediments which could be considered as its basement, but these were not metamorphic then.
Only in the OMZ, greenschist to amphibolite facies Neoproterozoic represents the Cadomian



basement, but it cannot be distinguished from the overlying Paleozoic metasediments in the NI
profiles. An even older crystalline basement of felsic composition exists, as indicated by
inherited zircons of 830-2000 Ma found in Ediacaran orthogneisses, Lower Ordovician
volcanics and Variscan granitoids that resulted from partial melting of such a basement
(Fernández-Suárez et al., 1998; Montero et al., 2007; Villaseca et al., 2012). Again, its upper
boundary is not imaged on NI profiles. These data also suggest that, in the Iberian Massif,
there are no crustal intervals that can be related with a seismic middle crust. Decoupling of
reflectivity, i.e. deformation, at a mid-crustal level define just an upper and a lower crust.
**4.5. Significance of a mid-crustal discontinuity: the Conrad discontinuity?**
Inspection of the Iberian Massif NI seismic dataset leads us to conclude that an orogenic-scale
mid-crustal discontinuity exits. This surface does not always provide a clear reflection, as in the
SPZ, but it is clearly defined by the geometry of the upper and lower crustal reflections,
asymptotically merging into it. The discontinuity coincides with the top of the lower crust,
which is often much more reflective than the upper crust. Furthermore, this discontinuity has
probably acted as a detachment for Variscan deformation in the hinterland of the orogen and
in the SPZ. However, in the CZ, the transition between upper and lower crust is poorly defined,
in accordance with the fact that its basement was not affected by Variscan tectonics. There, a
detachment level interpreted as the sole thrust of the thin-skinned wedge occurs above the
lower crust, and no deformation decoupling is identified above or below this feature.
Simancas et al. (2013) already described this discontinuity on the basis of the asymptotic
geometry of the SPZ faults towards the middle of the crust. These authors concluded that its
depth greatly varies when reaching suture boundaries, where the discontinuity roots. Although
we do not observe a subduction zone in the reworked elusive suture between the SPZ and the
OMZ (Pérez-Cáceres et al., 2015), and interpret the OMZ/CIZ suture as an indentation between
two continental crusts, triggering imbrication into the mantle of the latter (crocodile
structure), we agree that this discontinuity would have eased the decoupling of the Iberian
crust, allowing subduction of its lower part while the upper part was deformed by folds and
thrust faults. This is clearly observed in the Alpine northward subduction of the Iberian Massif
lower crust underneath the CZ (ESCIN-2, Fig. 4) and also, in the Pyrenees, where a detached
Iberian lower crust subducts to the N (Teixell et al., 2018). In the Iberian Massif, the complexity
of Variscan tectonics and late-Variscan crustal re-equilibration has mostly removed evidences
of a such mechanisms, but a comparable example has been preserved in the NW: the thick
lower crust imaged by ESCIN-3.3 is interpreted as underthrusting of the CZ lower crust under
that of the WALZ (Ayarza et al., 1998; Martínez Catalán et al., 2003, 2012, 2014).
The mid crustal discontinuity has been interpreted as the brittle-ductile transition (e.g. Ehsan
et al., 2014; Simancas et al., 2013). Indeed, it bounds a lower crust, highly reflective and
ductilely deformed from the upper crust. However, Variscan ductile deformation occurs also
above the discontinuity and is a general feature of the whole Iberian Massif except for the CZ.
If we deal with present deformation mechanisms, it is unlikely that the brittle-ductile
transition, which depends on the values of P and T, coincide with the described discontinuity,
because, i) it does not necessarily imply an impedance contrast (Litak and Brown, 1989), and
ii) according to figure 10, the depth of the discontinuity varies from 4 s TWT (~12 km, ALCUDIA

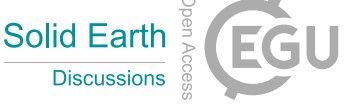

section) to 8 s TWT (~24 km, ESCIN-2 section) which would imply unrealistic variations on P
and T in the present crust.
The Iberian laminated lower crust is probably very old. Granulites dredged in Mesozoic
sediments of the Cantabrian margin have yielded ages of up to 1400 Ma (Capdevila et al., 1980
and references therein). Even older values have been obtained for granulites from the Galicia
Bank (Gardien et al., 2000), which featured Ar-Ar ages of up to 2500 Ma. These granulites have
been deformed ductilely during several orogenies. Rocks lying above the lower crust, whatever
their nature is, are separated from it by a discontinuity that fosters decoupled deformation
between both crustal layers. Accordingly, we think that the observed mid-crustal discontinuity
represents a rheological boundary that separates rocks that have been deformed differently.
The boundary, located at the top of the lower crust, represents a velocity contrast as the latter
is probably composed of dense granulites and includes relatively abundant basic rocks, which
makes it easily identifiable in seismic sections.
The geometry of this discontinuity and its depth, together with that of the Moho (Fig. 10),
provide insights of the evolution of the Iberian Massif. Along the SW Iberian Massif, the mid-
crustal discontinuity is sub-horizontal and lies at a depth between 4-6 s TWT. In the OMZ, the
intrusion of the IRB allows to establish its depth in the top or the bottom of this feature, but in
average, its location would fit the above given values. However, in the centre and NW, the
position of the discontinuity varies, deepening down to 8 s TWT (Figs. 9 and 10).
The low resolution noise autocorrelation models obtained along the CIMDEF profile shows
confusing results along the central Iberian Massif. In central Iberia, the mid-crustal
discontinuity might lie at 5-6 s TWT, deepening around the ICS to 8 s TWT as it has been
affected by pervasive extension and melting, thus defining a thin lower crust (2 s TWT, ~6 km,
Figs. 9 and 10). Accordingly, this feature appears redefined in this area, and now follows the
geometry of the ICS batholith. The change in the depth and geometry of this discontinuity and
the thinning of the lower crust might have allowed coupled deformation, letting part of the
upper crust to the S of the ICS to underthrust it (Andrés et al. this volume). This would foster
the 400-500 m topographic change between the N and S foreland basins of this Alpine
mountain range (Fig. 9). In fact, Simancas et al. (2013) argues that coupled crustal deformation
takes place when a relatively weak lower crust exists something that might well represent the
context of the ICS. The resulting geometry of this Alpine reactivation and its topographic
imprint is different to that observed to the N, in the CZ, where late orogenic extension and
melting does not exist and the mid-crustal discontinuity has been preserved. On the other
hand, the lower crust imaged along the CIMDEF transect presents a conspicuous internal
reflection that could also be interpreted as the top of the lower crust. If this were the case, the
lower crust would be even thinner along the entire section, matching the characteristics
observed to the N of the ALCUDIA section and in the ESCIN-3.2 line. In any case, we argue that
the mid-crustal discontinuity and the lower crust we are seeing in the CIMDEF profile are both
probably reworked by extension but not totally re-equilibrated and thus, its seismic image is
confusing. Moho depth models (Fig. 10) derived from shear wave tomography (Palomeras et
al., 2017) indicate that along the CIMDEF profile the crust is thin (except in the Alpine root) but
not as much as in NW Iberia, so that lower crustal extension and re-equilibration may have not
been as intense as in the GTMZ and CIZ of the NW Iberian Massif.





The most outstanding change in the mid-crustal discontinuity architecture appears in NW
Iberia along the ESCIN-3.2 profile. This section features the thinnest crust (9 s TWT)
accompanied by the thinnest lower crust (~1 s TWT). The mid-crustal discontinuity lies at 8 s
TWT in contrast to the depth where it appears in the neighbouring ESCIN-3-3 and ESCIN-1
lines, where it is located between 6 and 8 s TWT, suggesting that it has been redefined.
Nevertheless, clear reflections root in its upper part indicating that it still acted as a
discontinuity/detachment. The depth of this feature in the NW corner of Iberia is similar to
that of the high amplitude lower crustal internal reflection near the ICS. Accordingly, we
suggest that in NW Iberia, gravitational collapse followed by crustal melting and extension has
thinned the crust (Fig. 10), and specially the lower crust, relocating the mid-crustal
discontinuity.
NW Iberia was importantly thickened (up to 50-70 km) due to the emplacement of the GTMZ
allochtonous complexes. Thermal models by Alcock et al. (2009, 2015) show that as a result
the upper mantle continued increasing its temperature 60-65 Ma after the start of
compressional deformation at 360 Ma. This implies large thinning of the mantle lithosphere,
from 70 to 25-30 km, due to the ascent of the 1300 °C isotherm. It is not surprising that the
lower crust there became the most extended as a consequence of the heat increase, as in the
models it reached 800 °C after 45 Ma and 900 °C after 55 Ma (315-305 Ma).
The idea of a mid-crustal velocity discontinuity was put forward in the 1920's (Conrad, 1925).
Early analysis of natural source earthquake recordings and later images from controlled source
seismic reflection data provided further evidences that supported a clear distinction between
upper and lower crust. These evidences led to considering the Conrad discontinuity, a global
scale feature present the continental crust. However, this was later challenged as some results
of deep seismic reflection profiling did not show a clear distinction between upper and lower
crust (Litak and Brown, 1989).
Mid-crustal discontinuities have, however, been observed very often and in different types of
seismic data worldwide (e.g., Fianco et al., 2019; Hobbs et al., 2004; Melekhova et al., 2019;
Oncken, 1998; Ross et al., 2004; Snelson et al., 2013). Important changes in the rheology of the
crust have also been reported at those depths (Maggini and Caputo, 2020; Wever, 1989)
supporting the idea that a mechanical boundary must exist. Thus, we suggest that, even
though it is not observed everywhere (Litak and Brown, 1989), this feature is an orogen-scale,
world class crustal continental discontinuity (Artemieva, 2009), often coinciding with the top of
the highly laminated lower crust (when there is one). Its existence might determine the way
the crust deforms, easing decoupled deformation. Orogenic evolution, i.e. rifting, extension,
melting, etc. may modify it or even erase it, thus its existence and geometry might help us to
understand the geologic history of continents. In this regard, and coming back to the long-
forgotten discussion of the nature of the Conrad discontinuity (Conrad, 1925) and its position
on top of the laminated lower crust (Wever, 1989), we suggest that, in the Iberian Massif, the
observed mid-crustal feature fulfills the characteristics of this debated discontinuity. Its clear
signature and regional extension contributes to unravel its nature and significance.
**5. Conclusions**



Normal incidence seismic data acquired across the Iberian Massif in the last 30 years have
provided an entire section of a well exposed and almost complete part of the European
Variscides. Existing gaps in the central part have been recently sampled by passive source
seismic recordings (noise and earthquakes) that provide fairly good constraints on the crustal
structure.
Results show that crustal thickness varies from ~9 s TWT in late-Variscan extended areas (NW
of the Central Iberian Zone), to ~10 s TWT (30-33 km) in the external South Portuguese Zone to
~12 s TWT (36-38 km) in the internal West Asturian-Leonese Zone. Alpine reactivation has
managed to further thicken the crust to at least ~14 s TWT (42-45 km) in the external
Cantabrian Zone and to 35-38 km in the Iberian Central System, a Tertiary orogenic belt
developed in Central Spain. The top of a thick (up to 6 s TWT) and very reflective lower crust
helps to define a mid-crustal discontinuity across the entire Iberian Massif. This boundary
represents a level where reflections from the upper and lower crust merge asymptotically,
thus suggesting that it has often acted as a detachment or a decoupling level. Its position and
geometry varies mostly in relation to the late Variscan evolution. Accordingly, it is deeper in
NW and central Iberia (~8 s TWT), where Variscan crustal thickening was important and
gravitational collapse melted and extended the crust, thus defining a very thin lower crust.
However, it appears between 4-6 s TWT to the SW, where the crust did not thicken as much
and its original structure is better preserved, being later re-equilibrated through slow isostasy
and erosion.
This discontinuity exists in all the Iberian Massif tectonic zones, regardless of their Gondwana
or Avalonia affinity, thus suggesting it is an orogenic-scale discontinuity. We interpret it as the
rheological boundary between an overly ductilely deformed old lower crust and a
heterogeneous variably (often also ductilely) deformed upper crust that mostly (but not only)
shows evidences of the latest orogenic event. Its geometry, position and extent match the
characteristics defined for the long-forgotten Conrad discontinuity. The identification of similar
features in normal incidence profiles worldwide supports its inclusion as a major crustal
discontinuity.
**Acknowledgements**
The seismic data was reprocessed using the commercial seismic signal processing software
Claritas. Funding for this research was provided by the Junta de Castilla y León (SA065P17), the
Spanish Ministry of Science and Innovation (CGL2016-78560-P) and the Generalitat de
Catalunya (grant 2017SGR1022).

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

Table 1:  Acquisition parameters of the NI seismic profiles shown from figures 3 to 8 and
former processing flows. These are grouped according to their similarities.

| 1416    Acquisition parameters | ESCIN-1 (onshore) | ESCIN-2 (onshore) |
|---|---|---|
| 1417    Source | Dynamite-single-hole | Dynamite-single-hole |
| 1418    Charge | 20 kg at 24 m depth | 20 kg at 24 m depth |
| 1419    Trace Interval | 60 m | 60 m |
| 1420    # of traces | 240 | 240 |
| 1421    Spread configuration | Symmetrical Split-spread | Symmetrical Split-spread |
| 1422    Fold | 30 | 30 |
| 1423    Geophones per group | 18 | 18 |


| 1424 | Spread length | 14.5 km | 14.5 km |
| 1425 | Sample interval | | 4 ms |
| 1426 | | | |
| 1427 | | **ESCIN-3.2 and ESCIN-3.3 (offshore)** | |
| 1428 | | | |
| 1429 | Source | Airgun (5490 cu.in) | |
| 1430 | Shot spacing | 75 m | |
| 1431 | Receiver interval | 12.5 m | |
| 1432 | Spread length | 4500 m | |
| 1433 | Fold | 30 | |
| 1434 | Internal offset | 240 m | |
| 1435 | Sample rate | 4 ms | |
| 1436 | Record length | 20 s | |
| 1437 | | | |
| 1438 | | | |
| 1439 | | **IBERSEIS (onshore)** | **ALCUDIA (onshore)** |
| 1440 | | | |
| 1441 | Source | 4, 22T vibrators | 4 (+1), 22T vibrators |
| 1442 | Recording instrument | SERCEL 388, 10 Hz | SERCEL 388, 10 Hz |
| 1443 | # active channels | 240 minimum | 240 minimum |
| 1444 | Station spacing | 35 m | 35 m |
| 1445 | Station configuration | 12 geophones | 12 geophones |
| 1446 | Source spacing | 70 m | 70 m |
| 1447 | Sweep frequencies | non-linear 8-80 Hz | non-linear 8-80 Hz |
| 1448 | Sweep length | 20 s | 20 s |
| 1449 | Listening time | 40 s | 40 s |
| 1450 | Sample rate | 2 ms | 4 ms |
| 1451 | Spread type | Asymmetric split-spread | Asymmetric split-spread |
| 1452 | Nominal fold | 60 (minimum) | 60 (minimum) |
| 1453 | | | |
| 1454 | | | |
| 1455 | | | |
| 1456 | Figure captions | | |
| 1457 | | | |

Figure 1: Map of the Iberian Peninsula showing the outcrops of the Variscan basement and the
subdivision in zones of the Iberian Massif. The main strike-slip shear zones and gneiss domes
are included. Blue lines show the position of normal incidence seismic reflection profiles and
that of the CIMDEF transect. Abbreviations: Allochthonous complexes of NW Iberia: B:
Bragança; CO: Cabo Ortegal; M: Morais; MT: Malpica-Tui; O: Órdenes. Strike-slip shear zones:
BCSZ: Badajoz-Córdoba; DBSZ: Douro-Beira; JPSZ: Juzbado-Penalva; PTSZ: Porto-Tomar; SISZ:
Southern Iberian. See legend for other abbreviations. Traces of the main Variscan folds and the
Variscan granitoids are also included.
Figure 2: Processing flow carried over the SEG-Y original stack sections. This task was
geared to improve  and homogenize the resolution of the seismic images while creating
new migrated sections. See Martínez García, (2019) for further details.
Figure 3: Migrated section of the NI seismic profile ESCIN-1 (Fig. 1), without (a) and with
interpretation (b). A sketch of the most important features is presented in (c). CDP:





Common Depth Point. TWT: Two-way travel time. WALZ: West Asturian-Leonese Zone. CZ:
Cantabrian Zone.  The position of the Narcea Antiform is indicated.
Figure 4: Migrated section of the NI seismic profile ESCIN-2 (Fig. 1), without (a) and with
interpretation (b). A sketch of the most important features is presented in (c). CDP:
Common Depth Point. TWT: Two-way travel time. WALZ: West Asturian-Leonese Zone. CZ:
Cantabrian Zone. DB: Duero Basin.
Figure 5: Migrated section of the NI seismic profile ESCIN-3.3 (Fig. 1), without (a) and with
interpretation (b). A sketch of the most important features is presented in (c). CDP:
Common Depth Point. TWT: Two-way travel time. WALZ: West Asturian-Leonese Zone. CIZ:
Central Iberian Zone. The offshore projection of the Viveiro Fault is indicated.
Figure 6: Migrated section of the NI seismic profile ESCIN-3.2 (Fig. 1), without (a) and with
interpretation (b). A sketch of the most important features is presented in (c). CDP:
Common Depth Point. TWT: Two-way travel time. GTMZ: Galicia-Trás-os-Montes Zone.
Figure 7: Migrated section of the NI seismic profile ALCUDIA (Fig. 1), without (a) and with
interpretation (b). A sketch of the most important features is presented in (c). CDP:
Common Depth Point. TWT: Two-way travel time. CIZ: Central Iberian Zone. CU: Central
Unit.
Figure 8: Migrated section of the NI seismic profile IBERSEIS (Fig. 1), without (a) and with
interpretation (b). A sketch of the most important features is presented in (c). CDP:
Common Depth Point. TWT: Two-way travel time. CIZ: Central Iberian Zone. CU: Central
Unit. OMZ: Ossa-Morena Zone. RORS: Rheic ocean reworked suture. SPZ: South Portuguese
Zone.
Figure 9: Joint geological interpretation of all the seismic sections (normal incidence and
seismic noise) whose location is shown in figure 1. (a): ESCIN-1, ESCIN3-3 and ESCIN-3.2.
(b): ALCUDIA and IBERSEIS. (c): CIMDEF. Special attention should be paid to the depth and
geometry of the Moho and mid-crustal discontinuity. Alpine structures (i.e. crustal
thickening) appear in ESCIN-1 and in CIMDEF. The rest are Variscan features.
Figure 10: Map of the Moho depth as derived from tomography of shear waves (seismic
noise and earthquakes, Palomeras et al., 2017) with the projection of the seismic profiles
already shown in figure 1 and described along the text. A sketch of the geometry of the
main discontinuities (Moho and Conrad) is also shown.



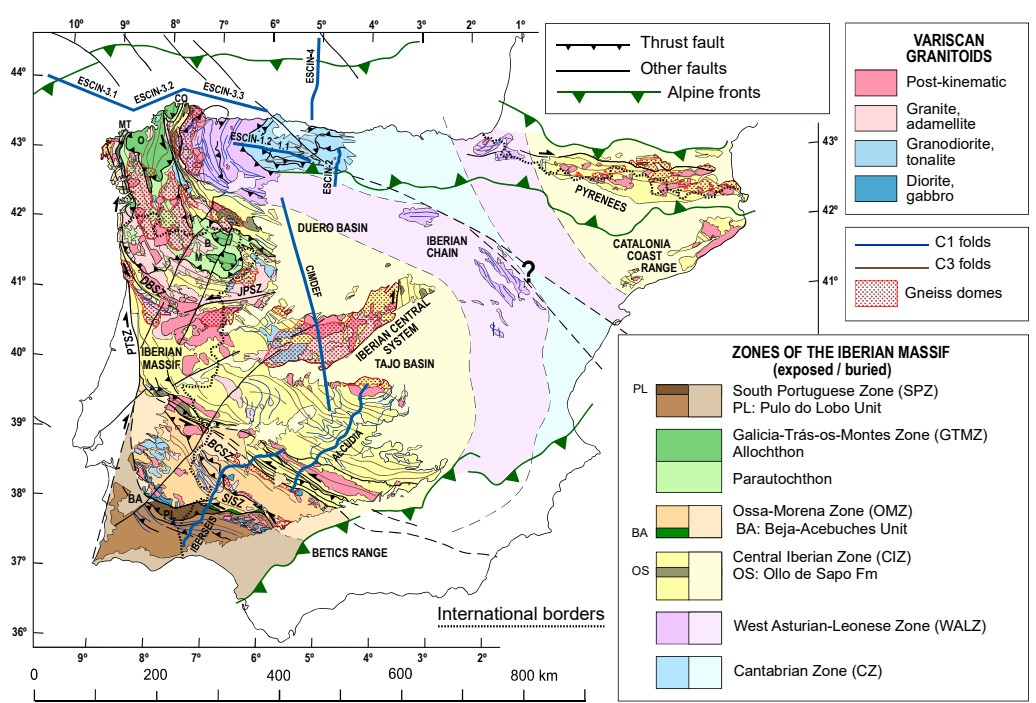

Figure 1



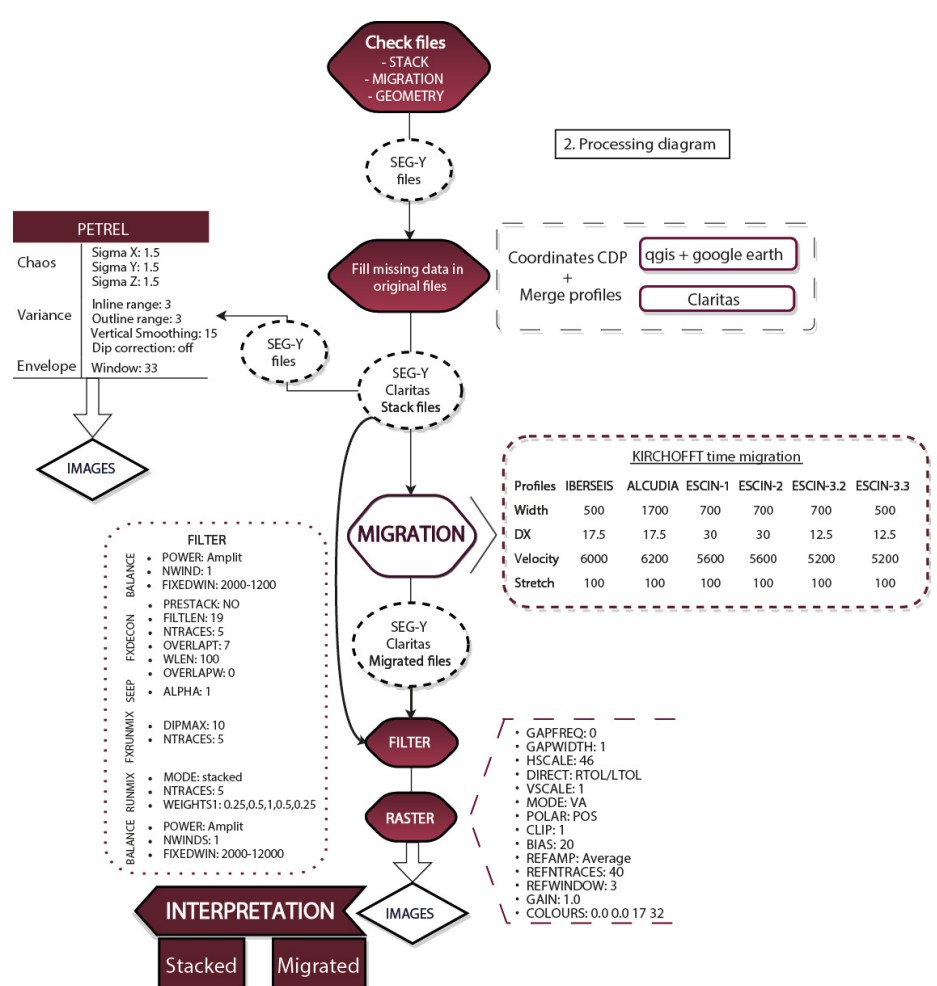

Figure 2




Figure 3

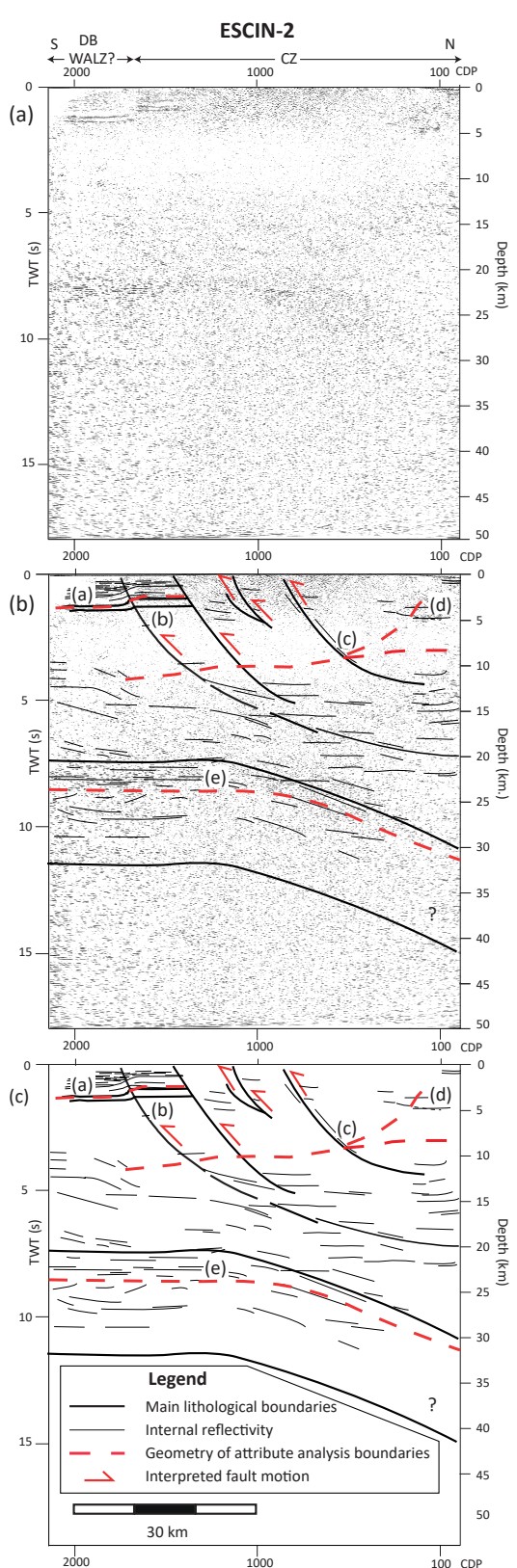

Figure 4





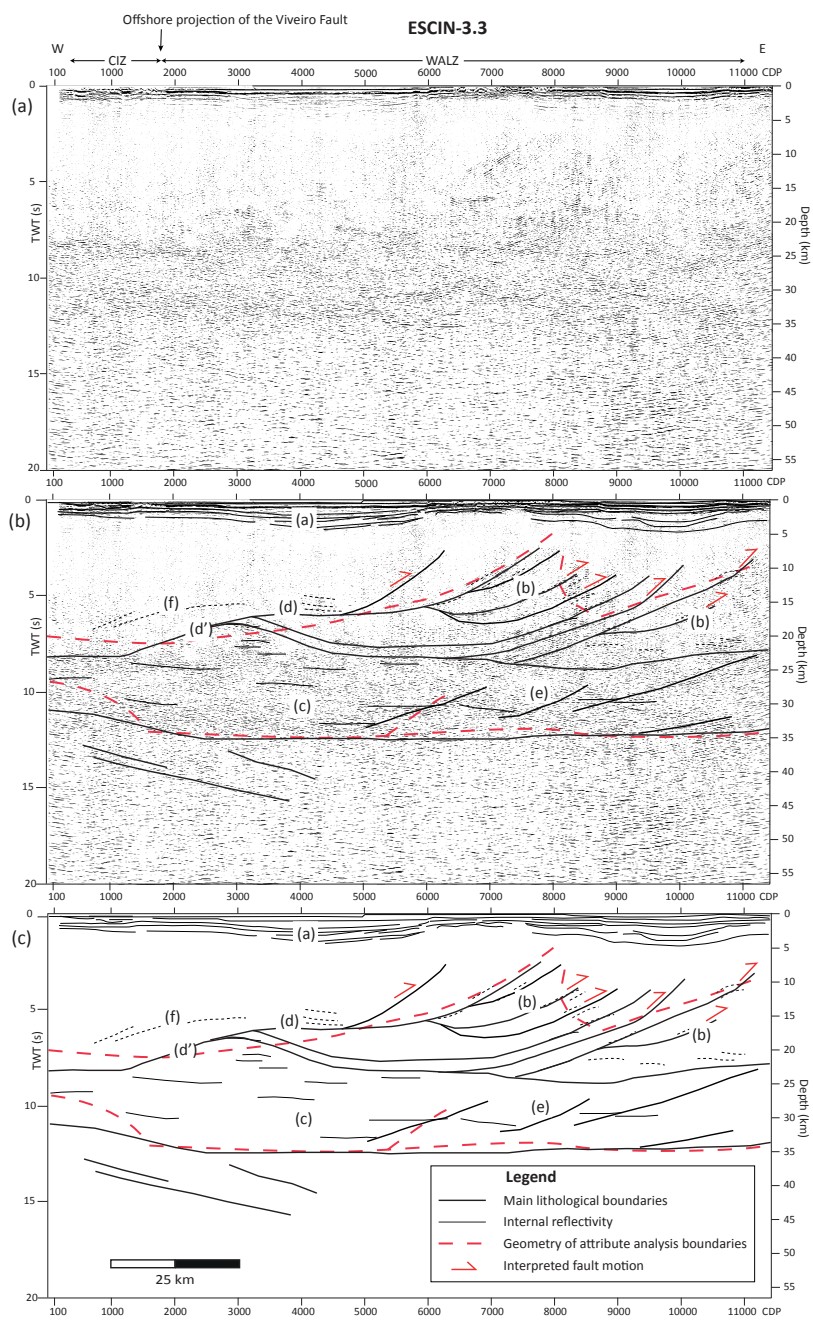

Figure 5



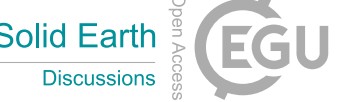

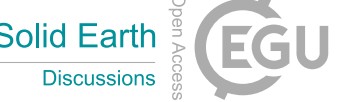

Figure 6





Figure 7



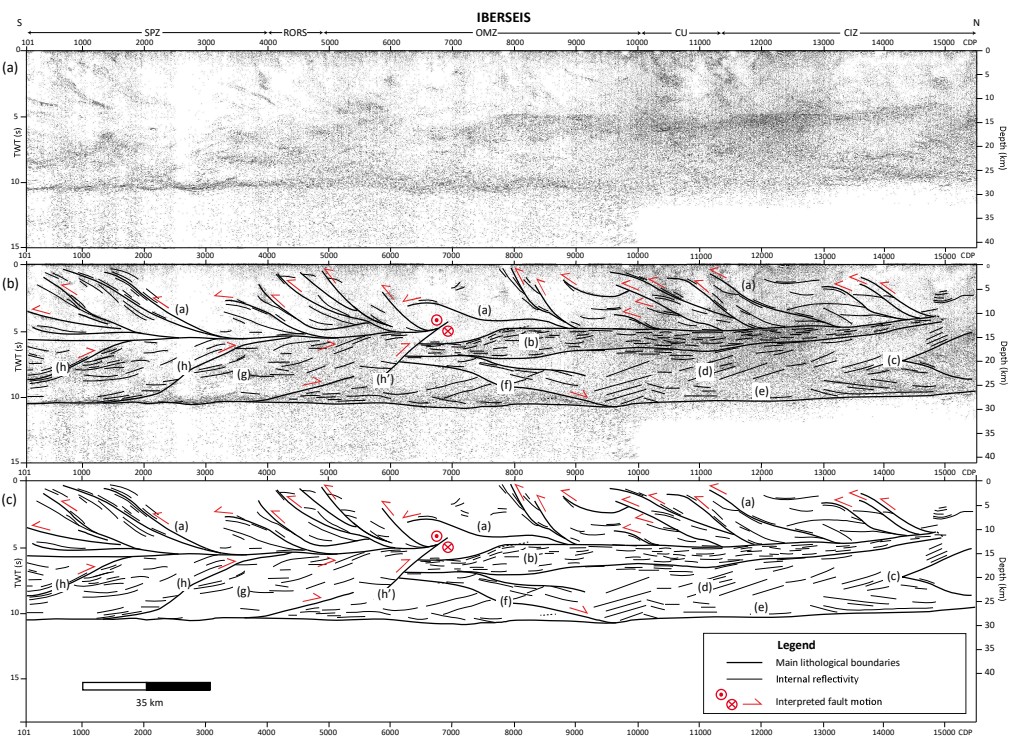

Figure 8



Figure 9
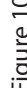

Figure 10