# Peer review of "Evolution of the Iberian Massif as deduced from its crustal thickness and 1"

_Solid Earth, 2020_

## Referee Comment (RC1) · Anonymous Referee #1 · 26 Nov 2020

Review of the paper: "**Evolution of the Iberian Massif as deduced from its crustal thickness and geometry of a mid-crustal (Conrad) discontinuity**"
by **Puy Ayarza, José Ramón Martínez Catalán, Ana Martínez García, Juan Alcalde, Juvenal Andrés, José Fernando Simancas, Immaculada Palomeras, David Martí, Irene DeFelipe, Chris Juhlin, Ramón Carbonell** submitted to Solid Earth open-access journal.

**Major comments**

The submitted manuscript concentrates on the reinterpretation of normal incidence seismic data along profiles in the Iberian Massif, Spain with the aim to present a cross-section through the Iberian Massif as the westernmost exposure of the European Variscides. The authors reprocess normal incidence data along 6 profiles and interpret them together with data along one other profile. They try to unify heterogeneity of the data during processing, which may have been lavish task to complete. Finally, they supplement these results with other result from the other profile CIMDEF.

In general, the paper is very extensive, discussing at length many geological evidences, however, the main goal of the paper is not clearly stated. The authors reprocessed 6 profiles though finally interpret 7 of them. About 6 reflection profiles, many of these data were already well documented and published (e.g. Simancas et al., 2013, Ehsan et al., 2014, etc.) documenting lots of details in the sections and interpretation and providing much more constraints than in the submitted manuscript. In the submitted manuscript, the authors show final reprocessed data, however, they do not provide any comparison with the previous results, they do not document the improvements or differences. So that, there is not clear evidence what is new in their submitted interpretation. Also, in the interpretation, they discuss details that are not constrained by the data in such a way. They also mention wide-angle data, but they do not provide velocity profiles though they convert TWT into depths which makes the interpretation less credible.

Except general comments, I have also difficulties with some interpretations provided in this manuscript. From Figs. 3-8, showing seismic sections and their interpretation, it is not clear how the interpretation is constrained, the interpretation is not correlated with geology, motion along lines (red arrows) is not documented, red dashed lines are not explained and do not correlate with the data, black lines termed Main lithological boundaries are not attributed to these lithologies, sometimes they are not constrained by the data (see e.g. line (d) in Fig. 3, bottom black line in Fig. 4b, line (c) in Fig. 4b, lines (b) in Fig. 5, numerous (a) lines in Fig. 8. Labelling of reflectors (a,b,c etc…) is not explained in figure captions and also sometimes does not correlated with the data. Interpretation is not the same for all profiles, see e.g. interpretation of data in Fig. 8 vs Fig. 7. (For more see also my comments below.)

According to two final figures and discussion, the authors seem to focus on two major discontinuities in the Iberian Massif – the Moho and the mid-crustal discontinuity. However, in my opinion, they are not well documented/constrained by the data at some parts (see, e.g., the bottom line in Fig. 4 in the N marked with "?" which does not seem to be justified by the data; or strong reflectors (d) at depths ~30-35 km in Fig. 6 considered by the authors as reflectors in the mantle with anomalously shallow Moho at depth 22km which may have natural also different explanation) and thus the interpretation might not be correct. The mid-crustal discontinuity is also discussed in relation to outdated term Conrad discontinuity, however, there is no comparison with velocity profiles which would help to discuss its genealogy. To conclude, I would suggest substantial reworking before potential resubmission.

**Further points:**

Figures in general – quality of figures in general is very poor. Tectonic setting is missing, seismic sections are poorly labelled, explanation of figures and labels is not provided in figure captions but only in text, geology is not provided in sections for comparison and references.

Fig. 1: needs reworking since there are difficulties to see profiles on top of tectonic structure. The geology is very detailed but the tectonic structure that is essential for the goal of this paper is not visible clearly from this figure.

Fig. 1: Tectonic zones referred in the text many times are not seen from Fig. 1

Interpretation of seismic sections along individual profiles:
Figs. 3-8 – Moho labelling should be provided in all Figs. 3-8; labels for individual horizons/reflectors should be provided in figure captions.

Figs. 3-8 in Legend – *Geometry of attribute analysis boundaries* – not clear what it means, it needs to be explained.

Fig. 3: I do not see subhorizontal reflector (e) extending through the whole section justified by the data.

Fig. 3: *Moho at 14.5s TWT* -> from Fig. 3 I can see the depth of ~40km, so what is correct?

Fig. 4: I do not see (b,c) structures interpreted in the section constrained by the data.

Fig. 4: It is not clear how the arrows are derived.

Fig. 4: I do not see the bottom line in the N marked with "?"justified by the data. How is it constrained? And is this Moho? If not, where is it?

Fig. 6: How can the authors derive the red arrows from the reflection data?

Fig. 6: *the Moho is located at around 9 s TWT (~27 km), the shallowest identified so far in the Iberian Massif* -> from Fig. 6 I can see strong reflector (c) at ~9s TWT, however, its depth is stated at 21-24 km. This seems to be very shallow if this is the Moho. This needs to be explained.

Fig. 6: strong reflectors (d) at depths ~30-35 km considered by the authors as reflectors in the mantle with anomalously shallow Moho at depth 22 km above. How can the authors distinguish what is the crust and what is the mantle? Cannot the (d) reflector at depths of ~30-35km represent the crust/mantle boundary? This needs to be explained.

Fig. 7: Red dashed lines – not clear what they represent and how they are related to the initial normal incidence data.

Fig. 7: *fairly transparent upper crust* -> I do not see fairly transparent upper crust throughout the whole section, it might be transparent in certain parts (e.g. around 6000-7000 CDP), however it exhibits variations aside. Also previous interpretations (Ehsan et al., 2014) show reflections in the upper crust along this section.

Fig. 8: I am surprised by the level of interpretation presented in Fig. 8 compared to e.g. Fig. 7. Are the data that much different? I cannot see so many reflectors from section in fig. 8, on the other hand, I can see more structure in the upper crust in section of fig. 7 (see also my comment above).

Fig. 8: red circles are not explained.

The authors include non-reprocessed profile CIMDEF in their final interpretations (see Fig. 9 and 10), however, they do not provide references nor even discuss this profile in respect to differences in the interpretation methods. Since the results along this profile were achieved from different data and methods, comparison of results in this respect is crutial.

Geology section is very long and comprehensive, however, it does not give the generalised overview of tectonic setting that is necessary for the goal of the paper (if it is the Moho and mid-crustal discontinuity).

192-198 – This para does not fit the overall text on tectonics.

299,338 – Is the migration velocity same for all profiles? And why there is chosen this value of 5600m/s?

Pfs ESCIN 3.3., 3.2., ALCUDIA and IBERSEIS do not state any migration velocity. It should be included.

264-283 processing of datasets – This chapter does not state, which data are to be processed and interpreted. This should be included.

326 – *Moho at 14.5s TWT* -> from Fig. 3 I can see the depth of ~40km, so what is correct?

424 – *the Moho is located at around 9 s TWT (~27 km), the shallowest identified so far in the Iberian Massif* -> from Fig. 6 I can see strong reflector (c) at ~9s TWT, however, its depth is at 21-24 km. This seems to be very shallow if this is the Moho. I can also see strong reflectors (d) beneath termed subcrustal reflectors. They are at depths of ~30-35km. How can the authors distinguish what is crust and what is the mantle? Cannot the (d) reflector at depths of ~30-35km represent the crust/mantle boundary?

448 – fairly transparent upper crust -> I do not see fairly transparent upper crust throughout the whole section, it might be transparent in certain parts (e.g. around 6000-7000 CDP), however it exhibits variations aside. Also previous interpretations (Ehsan et al., 2014) show reflections in the upper crust along this section.

There too many references (over 100, some of them are local in Spain and not accessible for broad readership) – needs elimination and restriction to only the most relevant ones.

**Minor edits**

I would suggest many minor edits, however, at this stage of reworking they are not relevant. Just few ones to state here:

43 – reference in missing

104, 139 cropping out -> outcropping

328, 696, 799, … and many other times – related with –> related to

709 – underplating

1425 – sample interval -> sample rate

1425 – sample rate is missing

1417-1425 – migration velocity is missing - stated only in text

1429-1452 – migration velocity is missing

Fig. 9 – profile CIMDEF is not interpreted –  reference is missing

Fig. 10 – what is the lilac dashed line?

Fig. 10 – profile CIMDEF is not re-interpreted, it is already published, however, reference is missing

Reference to figures needs to be unified throughout the manuscript. Sometimes they are referred as Fig., sometimes as Figure, sometimes as figure. It needs to be unified.

---

## Referee Comment (RC2) · Rob Butler (Referee) · 6 Jan 2021

This paper presents and interprets a composite seismic profile and that images the crust and upper mantle across the Iberian peninsular. It's an interesting topic: it is likely to have broad appeal beyond the immediate communities working on tectonic problems in this part of Europe. But as a contribution for the Special Issue - the topic is ideal. The authors are to be commended in showing the seismic images in both un-interpreted and interpreted form. However, it is to be hoped that the clean versions will be made available in large format, beyond the limitations of the standard publication. . . The manuscript is generally well-written – although there are a few glitches (some

of which are picked up in the points below). The use of information of WA seismic experiments – to better constrain velocities, and "normal incidence" records (for reflections) is commendable. It's good to see a well-documented seismic processing workflow. Discussion of the challenges of merging imagery from different experiments is rather brief (Section 3.2) but the points are well-made. This is a non-trivial task, especially given the experiments were in different tectonic domains, with distinctly different near-surface geology. This of course makes it difficult to know how much difference between adjacent profile segments is due to real structural variation and how much is associated with differences in acquisition and early processing. Are there descriptions of any sensitivity analysis in the various migration and attribute enhancements available? The authors describe the seismic images, interlaced with geological interpretation, sector by sector. I think the narrative would flow better if the seismic reflector patterns were described first and then interpreted. The interpretation of any seismic profile of course carries uncertainty – and this is especially true for imaging complex structures in non-sedimentary successions. The paper would benefit if the authors' preferred interpretation strategy and expectations were laid out explicitly at the outset. The introductory preamble is useful but maybe presents a rather optimistic view of the relationship between a seismic image and deformation structures in the crystalline crust. There are two distinct parts to this. The discussion assumes that the reflectivity in the, rather transparent, upper part of the profiles (less than 4-5 s TWT) is from faults/shear zones which therefore have a very simply form. This expectation is despite the complex geology and structural geology reported from outcrop. I think the interpretation of apparently continuous inclined (and locally apparently listric) reflectors in the top 5 seconds to be faults is at least open to debate. While cartoons of idealised imbricate thrust systems show such structures, they are pretty rare in my experience in nature! Second, the authors expect the continental crust to have a long-distance layered character with geophysically distinct "upper", "middle" and "lower" crust. Where this tri-partite structure is obscure in their images they infer "missing middle crust".... Of course there is middle crust present – there's not a void between deep crust and

upper crust layers! So presumably they mean that the interval between say 4-8 s TWT does not match their expectations. . ... Certainly, it is interesting that the transect shows a consistently reflective seismic "lower" crust (i.e. c 5-11 s TWT) – though it may be better to say that there is a consistently near-transparent shallow crust (1-5 s TWT). Personally I'd make more of the sub-Moho reflectors – perhaps referencing other such features imaged elsewhere in the world (e.g. the Flannan "event" in BIRPS images). If the authors are correct in their interpretation that the Iberian crust has been stacked by thrusts, then long-range layering might not be expected. . . unless it over-prints the Variscan structures. . ... in which case how much of the image relates to Variscan tectonics at all? The points made above indicate that I found the rolling discussions on the tectonic interpretation rather confusing. This may reflect the the difficulties in reconciling competing views amongst the extensive authorship! The Geological Setting notes are useful but quite involved, detailed and dense. The only illustration that accompanies the text is the geo-tectonic map of Iberia. As such it is very difficult to follow. How much of this do I, as a reader, need to retain to pick up the story...? For example, is the timing and delay of anatexis (line 132) really needed for the interpretation of the seismic data later? The message I get from the "Geological Setting" is that the structure of the present-day near surface is complex. . . including folds – that include deformed thrusts and thrust sheets (e.g. lines 150-157; line 187) – which is not conducive to their seismic imaging. . . For readers not familiar with region, some kind of palaeotectonic framework diagram could help to reinforcing the content. Likewise, some simple diagrams illustrating the competing models and interpretations of crustal structure would be useful – and these could then allow the seismic interpretations of the composite profile to be reframed as tests against these models.

Line 770 etc alludes to important ambiguities resulting from the interpretation of out-of-plane and migration artifacts. More could be made of this in discussion of interpretation uncertainties.

The interpretation is interwoven with basic description of seismic character. I think the

narrative would flow better if the seismic reflector patterns were described first and then interpreted. The narrative would benefit from a simple statement of assumptions and the preferred model at the outset (see above) – as much of the discussion here takes much of this as read. For example – line 461 and on states that the variations in the thickness (in TWT) of the reflective layer ("lower crust") imply differential thinning – extension. . .. But why? Could it not be that the reflectivity was developed hetero-geneously ? Or that the thicker portions have been thickened, rather than the thinner ones thinned?

Section 3.3 Is called a description of the seismic sections. It would be better indeed if this was what it was.. In fact, the section interlaces basic description of the seismic character with geological interpretation. In my view, the narrative would flow better if these two aspects were decoupled – so that first order description ("observations") are separated from the interpretation.

So describe reflection dips . . . Then say you infer that these track shear zone/thrust zone trajectories. Therefore where they go sub-horizontal then you deduce regional floor thrust positions.

Section 4.3 There are not many places in the world, away from Cenozoic orogens and basins, where continental crust is not underlain by a largely sub-horizontal Moho. Whether this represents gravitation flow of deep crust or simply differential isostatic rebound and concomitant erosion is debatable. Just how much upper crustal extension is there (stretching factors) from place to place? In settings like the Variscan – is the Moho a passive pre-orogenic marker – or is it a (partly magmatic or metamorphic) construct? There are interesting points in this discussions – many further references could be added. . .

Section 4.4 I found the premise here confusing. Metamorphic units are notoriously metastable – after all we get granulites and eclogites at the Earth's surface. Only if the metamorphism was in equilibrium and therefore over-printed previous assemblages

along modern (sub-horizontal) isotherms would the crustal seismic structure be as discussed here. But if so -the tectonic structure would (presumably) be hard to resolve – the intensities of reflectivity in the profiles could simply chart metamorphic (thermal) structure – not intensities of deformation as assumed here). …. You allude to this (line 761-2). But if so – when is the layering established? Presumably post-tectonically (after thermal re-equilibration)…

In the final discussion on the mid-crustal structure – description of the geophysical character is continuously intermingled with interpretation as a tectonic discontinuity. I would find it helpful if these two distinct aspects were separated. By all means set up the discussion in terms of Conrad – which is a geophysical construct. But make this distinct from its geo-tectonic interpretation.

Some detailed comments.

Personally, I find the continual use of acronyms distracting – especially short ones. It is easier for readers if you use Cantabrian Zone rather than CZ for example.

Line 57 – more complete than what? Better to say Our aim here is the present a composite seismic profile that integrates results from two new experiments (IMDEF and ALCUDIA WA) with existing data-sets (specify).

Line 60 – "Later on" makes it sound like it is another paper. "Here we continue to…" or some such might be clearer… continuing…We revisit interpretations of crustal extension and a possible mid-crustal discontinuity. We discuss mid-crustal reflectivity, the so-called "Conrad Discontinuity" of classical continental seismology (Conrad 1921), in the light of long-running debates as to its tectonic significance (REFS).

Line 87 – strictly the correlation does "support" the affinity – it is consistent with it …

Line 88 etc "Evidences" - the plural of evidence in this context is "evidence" (no "s", like sheep).

Line 95 – "in the surface" – do you mean at outcrop?

Line 98 – what is "it"? The structure of the Iberian Massif along a N-S transect. . .?

Line 139 and Line 140 etc. Be consistent with the verb. . .– is it "cop out" or "outcropping"..

Line 183-184. Statements like this are key. . . mid crustal reflectivity can be explained by intrusions. . .. But what evidence is there that they were controlled by shear zones? Why does reflectivity necessarily track deformation?

Line 229 etc. a plural of a date has no apostrophe – it's 1990s. . ..

Line 237 – kind of experiment (no need for plural).

Line 285-287. Please reference explicitly these primary sources for the seismic processing. Hopefully these are peer-reviewed, formal publications!

Line 305 (and many other places). Interpretation is presented as fact. So "W-dipping reflections that represent the Variscan imbrication" – is highly interpretative. First it would help if this statement is justified. . .. How explicitly does the reflectivity match to outcrop structure?

Line 312 – Interesting – but when thin-skinned interpretations were provided by (eg) COCORP Appalachians from 1970s– they tied reflectivity to underthrust sediments that could be traced down from outcrop. . .

Line 448 etc. I'd avoid using the phase "is related to" when discussing the seismic expression with respect to the surface geology. A better basic phrase is – "coincides with" – as this avoids associating description with interpreted causation. . .

Line 459 – Can you exclude the "cross-cutting" relationships are in-plane migration (or out of plane) artifacts. . .?

Line 477 "Mantle" reflectivity – what evidence is there to support the notion of crust-mantle imbrication? Could this not be intra-mantle structure?

Line 707 – which author? Meissner??

---

## Author Comment (AC1) · 5 Feb 2021

Response to reviews and comments on the MS: **"Evolution of the Iberian Massif as deduced from its crustal thickness and geometry of a mid-crustal (Conrad) discontinuity"**, by Ayarza et al.

Ahead, we provide answers to the comments on our paper presented by reviewer 1.

**Reviewer 1: Anonymous**

> …..however, the main goal of the paper is not clearly stated. The authors reprocessed 6 profiles though finally interpret 7 of them. About 6 reflection profiles, many of these data were already well documented and published (e.g. Simancas et al., 2013, Ehsan et al., 2014, etc.) documenting lots of details in the sections and interpretation and providing much more constraints than in the submitted manuscript. In the submitted manuscript, the authors show final reprocessed data, however, they do not provide any comparison with the previous results, they do not document the improvements or differences. So that, there is not clear evidence what is new in their submitted interpretation.

The goal of the paper is to provide a joint interpretation of all the seismic datasets acquired in the Iberian Massif. Up to date, they have been thoroughly interpreted individually, as the reviewer points out. However, there are features that need to be re-interpreted as the most recent models for the evolution of the Iberian Massif emphasize the importance of late orogenic extension in the configuration of the orogeny and its later reactivation during the Alpine Orogeny. This effect can be evaluated from the geometry of a mid-crustal discontinuity and that of the Moho. The upper crust interpretation has not significantly varied. Those features that were previously correlated with upper crustal faults and thrusts remain unmodified. It is just finally, in this paper, we stablish a correlation between this orogeny scale mid-crustal discontinuity and the Conrad discontinuity, thus providing a new definition regarding what this discontinuity might represent. According to the above, the introduction will be modified.

> Also, in the interpretation, they discuss details that are not constrained by the data in such a way. They also mention wide-angle data, but they do not provide velocity profiles though they convert TWT into depths which makes the interpretation less credible.

The details that are not constrained by the data, like the direction of movement of thrusts and faults, have been previously integrated with surface geology, interpreted and published. This is known by the reviewer, as acknowledge in the second paragraph of the comments, partially reproduced above, in the first "squared" comment. In our text, we provide cites of all the previous interpretations of the seismic profiles we present. It is not efficient to "...discuss details that are not constrained by the data..." since data are published elsewhere and references are provided. However, and also in relation with comments by the second reviewer, we will include in this version cross sections at the top of the seismic profiles to facilitate upper crustal correlations between geology and seismics. We will also refer to cites more often. Migration velocities presented in figure 2 and are calculated from wide-angle data and those are the ones used for depth conversion. This information will be added to the paper

> From Figs. 3-8, showing seismic sections and their interpretation, it is not clear how the interpretation is constrained, the interpretation is not correlated with geology, motion along lines

(red arrows) is not documented, red dashed lines are not explained and do not correlate with the data, black lines termed Main lithological boundaries are not attributed to these lithologies, sometimes they are not constrained by the data (see e.g. line (d) in Fig. 3, bottom black line in Fig. 4b, line (c) in Fig. 4b, lines (b) in Fig. 5, numerous (a) lines in Fig. 8. Labelling of reflectors (a,b,c etc…) is not explained in figure captions and also sometimes does not correlated with the data. Interpretation is not the same for all profiles, see e.g. interpretation of data in Fig. 8 vs Fig. 7.

As stated above, cross-sections are going to be added on top of seismic sections, and cites of the papers where the interpretation is made will be included in the figure captions. The red dashed lines are referred in the legend of figures as "Geometry of attribute analysis boundaries". Attribute analysis is one of the routines applied during reprocessing of the profiles, as explained in section 3.2: Processing of datasets, lines 277-283, and described in two references cited (Chopra and Alexeev, 2005; Taner and Sheriff, 1997). However, we acknowledge that the non-specialized reader may ignore what these routines do, so that a brief explanation will be added. Finally, red dashed lines are expected to discriminate between different types of crust (lithologies) at a large scale. Even though sometimes they do not have exactly the same geometry as reflections, we prefer keeping them as they often add constraints to the interpretation.

Line d in figure 3 marks the top of the lower crust. Bottom black line in figure 4b is constrained in the southern part of the figure, which provides a lower crustal thickness. The rest is extrapolated from the geometry of the upper crust and previous publications. C in figure 4b is constrained by reflectivity at ~3 s TWT and surface geology. Lines b in figure 5 feature high reflectivity and their identification is straightforward. Finally, 'a' lines in figure 8 are well observed. Anyway, we will revise the identified reflections on the light of the cross-sections we are going to add. Finally, labelling description will be also included in the figure captions. In addition, we will change the name of the labels so the same features have always the same label (e.g., Moho_m)

According to two final figures and discussion, the authors seem to focus on two major discontinuities in the Iberian Massif – the Moho and the mid-crustal discontinuity. However, in my opinion, they are not well documented/constrained by the data at some parts (see, e.g., the bottom line in Fig. 4 in the N marked with "?" which does not seem to be justified by the data; or strong reflectors (d) at depths ~30-35 km in Fig. 6 considered by the authors as reflectors in the mantle with anomalously shallow Moho at depth 22km which may have natural also different explanation) and thus the interpretation might not be correct

As explained above, bottom black line in figure 4b is constrained in the southern part of the figure, which already provides a lower crustal thickness. The rest is extrapolated from the geometry of the upper crust and previous publications. Also the stack indicates that the geometry is similar to the one depicted but migration blurs the final image. There is an entire paper devoted to reflections d (Ayarza et al., 2004), where these have been modeled in 3D. Their origin and interpretation play no role in the scope of this paper as they are 3D alpine features coming from the southward subduction of the oceanic crust of the Bay of Biscay. Of course, their origin might be different. But the only published model, so far, is that of Ayarza t al. (2004).

The mid-crustal discontinuity is also discussed in relation to outdated term Conrad discontinuity, however, there is no comparison with velocity profiles which would help to discuss its genealogy.

One of the points of the paper is to define the nature of the Conrad discontinuity as we observe it in an orogeny scale profile, with zones previously belonging to two supercontinenets (Gondwana and Laurussia). Velocity profiles derived from wide angle data are going to be included in some of the sections to help on this interpretation

Figures: Tectonic setting is missing, seismic sections are poorly labelled, explanation of figures and labels is not provided in figure captions but only in text, geology is not provided in sections for comparison and references.

Captions presently explain the content of the figures, without describing their main features and geological interpretation. For description and interpretation, the reader is referred to the text in an implicit way, that is, without adding the classical "See text for explanation" tag. However, cross sections will be added on top of the profiles. Labelling will be modified to make it more coherent regarding the whole of figures and will be included in the figure captions.

Fig 1 needs reworking since there are difficulties to see profiles on top of tectonic structure. The geology is very detailed but the tectonic structure that is essential for the goal of this paper is not visible clearly from this figure.

The main problem to see the tectonic structure in Figure 1 is the granitoids, but they are important as indicators of deep crustal processes. So, the figure will be duplicated (parts a and b), leaving the structure in one map and the granitoids in the other. The seismic lines will be highlighted changing their color if necessary.

Fig. 1: Tectonic zones referred in the text many times are not seen from Fig. 1

References to tectonic zones in the text will be revised and depending on their importance, removed, added in the figure or referred to their publication

Figs. 3-8 – Moho labelling should be provided in all Figs. 3-8; labels for individual horizons/reflectors should be provided in figure captions.

Moho labelling will be included in every figure. Furthermore, reflector labels will be included in figure captions and will be modified so they represent the same features in every profile.

Figs. 3-8 in Legend – Geometry of attribute analysis boundaries – not clear what it means, it needs to be explained.

More details about the attribute analysis will be given in the text

Fig. 3: I do not see subhorizontal reflector (e) extending through the whole section justified by the data.

Certainly reflector e is only well constrained in the stack. In the migrated section is only visible between CDP's 1000 and 2000. We will mark the rest with a dashed line and refer to the publication where the stack was interpreted

Fig. 3: Moho at 14.5s TWT -> from Fig. 3 I can see the depth of ~40km, so what is correct?

Depth conversion has been made using the migration velocities, which have been extracted from coincident or neighboring wide angle data. In the ESCIN-1 dataset, the migration velocity is 5600 m/s. So 14.5 s TWT converts to 40.6 km. The same criteria applies to the rest of the profiles

Fig. 4: I do not see (b,c) structures interpreted in the section constrained by the data

b reflections clearly offset sediments from the Duero Basin and c reflection can be identified from CDP 1000 to 500 above 7 s TWT. Certainly it loses reflectivity after migration but it can be followed and interpreted also, when compared with the published stack. Reference to this publication will be added in the description

Fig. 4: It is not clear how the arrows are derived

Cites and cross sections will be added so the sense of movement interpreted from the reflections is clear. But, this is an area affected by Alpine compressional tectonics, and the main faults are identified on surface.

Fig. 4: I do not see the bottom line in the N marked with "?"justified by the data. How is it constrained? And is this Moho? If not, where is it?

Yes, that reflection represents the Moho and, it is constrained by reflectivity at the southern part of the profile (which provides a lower crustal thickness). The rest is extrapolated from the geometry of the upper crust and previous publications based on the stack image

Fig. 6: How can the authors derive the red arrows from the reflection data?

In general, we derived them from the knowledge of the surface geology, the geometry of reflections and also assumed results from previous publications, and cross sections will be included in the figures to ease the reader the understanding of these features. In the case of this figure, the sense is not clear, as the reflections are deep and cannot be correlated to structures at the surface. It images a part of the Variscan crust that was subjected to compression and thickening followed by very important extension. So, the most probable late-Variscan motion is to the west, and the possibility of reactivation of early crustal-scale thrusts should not be ruled out. The text has already a discussion regarding this issue (lines 429-437).

Fig. 6: the Moho is located at around 9 s TWT (~27 km), the shallowest identified so far in the Iberian Massif -> from Fig. 6 I can see strong reflector (c) at ~9s TWT, however, its depth is stated at 21-24 km. This seems to be very shallow if this is the Moho. This needs to be explained.

The ESCIN3-2 line is an offshore profile sampling sedimentary basins and featuring a thin crust. Thus, the migration velocity, derived from neighboring wide angle data is low: 5.2 km/s (Fig. 2), which implies that 8-9 s TWT convert to 20.8-23.4 km

Fig. 6: strong reflectors (d) at depths ~30-35 km considered by the authors as reflectors in the mantle with anomalously shallow Moho at depth 22 km above. How can the authors distinguish what is the crust and what is the mantle? Cannot the (d) reflector at depths of ~30-35km represent the crust/mantle boundary? This needs to be explained.

Yes, indeed. The narrow, sub-horizontal and highly reflective band between 8-9 s in ESCIN-3.2 is the continuation of the profile ESCIN 3.3 lower crust. Accordingly, the one strongly dipping to the W could be the continuation of another reflective band found in ESCIN-3.3 between 10-12 s and interpreted as a partial (only the basement) crustal duplication: the underthrusting of the Cantabrian Zone basement. We have explained this structure in several publications (e.g., Ayarza et al., 1998). The deep west-dipping reflections on Fig. 6 would be the continuation into de mantle of this partial crustal duplication. Considering the reviewers opinion, we will priorize this interpretation over the one provided in the first version of the paper and we will add a more thorough discussion, together with the pertinent references

Fig. 7: Red dashed lines – not clear what they represent and how they are related to the initial normal incidence data.

These lines represent the boundaries of the attribute analysis and roughly represent different types of crust/lithologies. More information about the attribute analysis is going to be added

Fig. 7: fairly transparent upper crust -> I do not see fairly transparent upper crust throughout the whole section, it might be transparent in certain parts (e.g. around 6000-7000 CDP), however it exhibits variations aside. Also previous interpretations (Ehsan et al., 2014) show reflections in the upper crust along this section.

The upper crust in this section is fairly transparent when compared with the IBERSEIS section. The reason is the abundance of vertical folds affecting little impedance contrast layers (lines….). Also, the existence of granitoids decreases reflectivity. However, it is true that reflectivity exists. Furthermore, reviewer 2 pointed out that the upper crust is certainly quite transparent. Nevertheless, we will rephrase the main description of this part of the sections

Fig. 8: I am surprised by the level of interpretation presented in Fig. 8 compared to e.g. Fig. 7. Are the data that much different? I cannot see so many reflectors from section in fig. 8, on the other hand, I can see more structure in the upper crust in section of fig. 7 (see also my comment above).

IBERSEIS shows the most reflective upper crust of the Iberian Massif so far. Also, the quality and continuity of the surface outcrops allows interpretation to reach deeper levels. Again, we will rephrase some of the sentences used in the description of these two profiles.

Fig. 8: red circles are not explained.

Symbols with red circles show possible fault motion, as indicated in the legend. In this case, they indicate sinistral wrench motion. Of course, we ignore the real motion of the fault traced on the profile, and the symbols only try to represent in some way the generalized sinistral transcurrence at the boundary between Ossa-Morena and South Portuguese zones. They can be deleted or further details can be added in the figure caption.

The authors include non-reprocessed profile CIMDEF in their final interpretations (see Fig. 9 and 10), however, they do not provide references nor even discuss this profile in respect to differences in the interpretation methods. Since the results along this profile were achieved from different data and methods, comparison of results in this respect is crutial.

Indeed, more details about this dataset will be added. At the time of the submission of this paper, the CIMDEF dataset was under minor revision but it is now published. So now, cites referring to its publication will be added. However, keep in mind that this is not a vertical incidence profile but it has been obtained by interferometry of natural source data (noise). These type of data is geared to show sub-horizontal reflections and thus, time migration does not help in the interpretation. Authors publishing that dataset have not deliberately migrated it and neither have us. Its resolution is also lower (2.5-4 Hz) so CIMDEF is just used to constrain the depth and geometry of the mid-crustal detachment and the Moho in the area where a vertical incidence gap exists.

> Geology section is very long and comprehensive, however, it does not give the generalised overview of tectonic setting that is necessary for the goal of the paper (if it is the Moho and mid-crustal discontinuity).

Does the reviewer refer to the Geological Setting section? We will include more information that can be useful in the understanding of the Moho depth according to geology. But the discovery of the mid-crustal detachment has been only possible after the analysis of vertical incidence sections so we cannot include it in the geological setting

> 192-198 – This para does not fit the overall text on tectonics.

Again, does the reviewer refer to the Geological Setting section? There, an overview of the tectonics of the Iberian Massif is done between lines 66-99, and the Carboniferous extension and mafic intrusions are mentioned on lines 82-87. Furthermore, the IRB is probably one of the most outstanding features identified in the seismic data of the Iberian Massif. Although including seismic data interpretation in the Tectonics section is not desirable, the implications of this body in the support of the Early Carboniferous extension mnade us to include it, and we prefer to keep it.

> 299,338 – Is the migration velocity same for all profiles? And why there is chosen this value of 5600m/s?

Figure 2 shows the migration velocities for every profile. They are different and have been chosen in relation with the wide angle velocities of overlapping or neighboring profiles.

> Pfs ESCIN 3.3., 3.2., ALCUDIA and IBERSEIS do not state any migration velocity. It should be included

Figure 2 includes the migration velocity of every seismic line presented. We will include this information in figure captions too.

> 264-283 processing of datasets – This chapter does not state, which data are to be processed and interpreted. This should be included.

We will include that

> 326 – Moho at 14.5s TWT -> from Fig. 3 I can see the depth of ~40km, so what is correct?

Depth conversion velocity is the migration velocity, which is based on wide angle data. In this case v=5600 m/s so the conversion is correct

424 – the Moho is located at around 9 s TWT (~27 km), the shallowest identified so far in the Iberian Massif -> from Fig. 6 I can see strong reflector (c) at ~9s TWT, however, its depth is at 21-24 km. This seems to be very shallow if this is the Moho.  I can also see strong reflectors (d) beneath termed subcrustal reflectors. They are at depths of ~30-35km. How can the authors distinguish what is crust and what is the mantle? Cannot the (d) reflector at depths of ~30-35km represent the crust/mantle boundary?

As it has been discussed elsewhere, depth conversion velocity is the migration velocity, which is 5.2 km/s for this profile. In addition, an as it has been discussed above, the high dipping features somehow represent a complex Alpine crust-mantle boundary (mantle-oceanic crust-mantle sequence) imaged in 3D, i.e, located to the N of the profile. As stated above, there is an entire paper devoted to reflections 'd' (Ayarza et al., 2004), where these have been modeled in 3D. Their origin and interpretation play no role in the scope of this paper as they are 3D alpine features coming from the southward subduction of the oceanic crust of the Bay of Biscay. Therefore, we consider that, other than citing the corresponding papers, there is no point in extending the information regarding those features.

448 – fairly transparent upper crust -> I do not see fairly transparent upper crust throughout the whole section, it might be transparent in certain parts (e.g. around 6000-7000 CDP), however it exhibits variations aside. Also previous interpretations (Ehsan et al., 2014) show reflections in the upper crust along this section.

In fact, that upper crust is very transparent as it is composed mostly of vertical folds affecting little reflective lithologies. Reflectivity is associated to few existing contrast between metasediments, granites and a suture zone. Furthermore, the second reviwer acknowledges that the upper crust in very transparent in general in the shown sections.

There too many references (over 100, some of them are local in Spain and not accessible for broad readership) – needs elimination and restriction to only the most relevant ones.

Some non-essential references of local character will be removed

43 – reference in missing

We will add it

104, 139 cropping out -> outcropping

We will change it

328, 696, 799, … and many other times – related with –> related to

We will change it

709 – underplating

We will change it

| 1425 – sample interval -> sample rate |
| --- |

We will change it

| 1425 – sample rate is missing |
| --- |

It will be added

| 1417-1425 – migration velocity is missing - stated only in text |
| --- |

It will be added

| 1429-1452 – migration velocity is missing |
| --- |

It will be added

| Fig. 9 – profile CIMDEF is not interpreted – reference is missing |
| --- |

It will be added

| Fig. 10 – what is the lilac dashed line? |
| --- |

Yes, our mistake. It is an intra-lower crust highly reflective feature, tentatively related with the extension by-product. It will be explain in the figure.

| Fig. 10 – profile CIMDEF is not re-interpreted, it is already published, however, reference is missing |
| --- |

This profile was under review at the time of the submission. Now, that it is accepted, the reference will be added

| Reference to figures needs to be unified throughout the manuscript. Sometimes they are referred as Fig., sometimes as Figure, sometimes as figure. It needs to be unified. |
| --- |

When in parenthesis we use Fig. Along the text figure or Figure depending if there is a period before. We will check every case and correct them.

---

## Author Comment (AC2) · 5 Feb 2021

Response to reviews and comments on the MS: **"Evolution of the Iberian Massif as deduced from its crustal thickness and geometry of a mid-crustal (Conrad) discontinuity"**, by Ayarza et al.

**Reviewer 2: Prof. Rob Butler**

Ahead, we provide answers to the comments on our paper presented by reviewer 2.

> Are there descriptions of any sensitivity analysis in the various migration and attribute enhancements available?

Not at this point

> I think the narrative would flow better if the seismic reflector patterns were described first and then interpreted.

Although that is our usual way of presenting and discussing vertical incidence seismic profiles, the high number of datasets presented in this paper and the fact that they have all been previously interpreted geared us to find a more agile way of presenting the data and we prefer to keep it this way. However, we are going to add more information in the figure captions and modify labelling so it is more uniform and valid for all figures (e.g., M: Moho, C: Mid-crustal Discontinuity (Conrad)). In addition, labels will be described in figure captions as well..

> The paper would benefit if the authors' preferred interpretation strategy and expectations were laid out explicitly at the outset. The introductory preamble is useful but maybe presents a rather optimistic view of the relationship between a seismic image and deformation structures in the crystalline crust.

This paper tries to make a joint interpretation of all Iberian Massif vertical incidence datasets acquired to date. Where there is a gap, seismic interferometry based on natural source seismic data (noise) is used. The seismic upper crust of every profile has already been interpreted and correlated with surface geology. Cites are given. Our contribution is to interpret the entire dataset on the light of new models that emphasize 1) the lateral extent of late Variscan extensional tectonics, identified after 2003 and which relevance is being found on field and geophysical data 2) the depth extent at which crustal melting has taken place in the NW quadrant of the Iberian Massif 3) the effect that this late tectonic processes had in the Alpine reworking of the Iberian crust. This task is made through the study of the geometry of two interfaces: 1) a mid-crustal detachment, addressed to the Conrad discontinuity, whose character is later discussed and 2) the Moho. The introduction will be modified so the goal of this paper is clearer.

> There are two distinct parts to this. The discussion assumes that the reflectivity in the, rather transparent, upper part of the profiles (less than 4-5 s TWT) is from faults/shear zones which therefore have a very simply form. This expectation is despite the complex geology and structural geology reported from outcrop. I think the interpretation of apparently continuous inclined (and locally apparently listric) reflectors in the top 5 seconds to be faults is at least open to debate. While cartoons of idealised imbricate thrust systems show such structures, they are pretty rare in my experience in nature!

We agree with the reviewer on the fact that the geometry of faults/thrusts is complicated. It is also our personal opinion that trusts themselves do not need to be very reflective unless they follow a detachment level characterized by contrasting lithologies. But the surface correlation between reflectors and mapped thrusts forced us, and cited authors, to interpret the former as thrusts. Besides, they have the geometry given for these features in cartoons. Finally, all sections are migrated, so we are reasonably confident about the geometry of reflections. If listric reflectors are not faults/thrust, what could they be instead? The reflectivity itself might be produced by contrasting lithologies, but in the Iberian Massif, mappable thrusts (detachments) are often identified to coincide with important lithological boundaries.

Second, the authors expect the continental crust to have a long-distance layered character with geophysically distinct "upper", "middle" and "lower" crust. Where this tripartite structure is obscure in their images they infer "missing middle crust". Of course there is middle crust present – there's not a void between deep crust and upper crust layers! So presumably they mean that the interval between say 4-8 s TWT does not match their expectations.

From a seismic point of view what we and most of the authors working in these data have seen is reflectors in the 'seismic' upper and lower crust merging in a mid-crustal discontinuity. Of course there is a middle crust, and there is a part in the discussion referring to what a metamorphic middle crust means. But these petrologically-speaking mid crustal rocks and even lower crustal rocks are, sometimes, emplaced in the surface by thrusts and/or extensional faults. Thus we find that seismically there is only upper and lower crust separated by a discontinuity often coinciding with a detachment. Furthermore, between 4 and 8 s TWT sometimes we find a layered and very reflective lower crust (ALCUDIA profile) and sometimes a fairly transparent upper crust (ESCIN-3.2 profile). So it is not a matter of depth or travel time. It is a matter of rheology of the crust and how the later accommodates deformation.

Certainly, it is interesting that the transect shows a consistently reflective seismic "lower" crust (i.e. c 5-11 s TWT) – though it may be better to say that there is a consistently near-transparent shallow crust (1-5 s TWT).

Surprisingly, reviewer 1 argues that the upper crust is not very transparent in the ALCUDIA section, where hardly any upper crustal reflections are found. However, we agree with reviewer 2 in interpreting a fairly transparent upper crust.

Personally I'd make more of the sub-Moho reflectors – perhaps referencing other such features imaged elsewhere in the world (e.g. the Flannan "event" in BIRPS images). If the authors are correct in their interpretation that the Iberian crust has been stacked by thrusts, then long-range layering might not be expected. . . unless it over-prints the Variscan structures. . .. in which case how much of the image relates to Variscan tec-tonics at all?

In our opinion, lower crustal layering is partly pre-Variscan, and partly overprinted by Variscan structures, especially by late-Variscan extensional ones. Sub-Moho reflections represent, in some cases, preserved Variscan crust/mantle imbrications. Their interpretation does not represent the main goal of the paper, but following this comment and another one made by reviewer 1, they will be dealt with in the revised version. In this regard, W dipping mantle reflections found in profile

ESCIN3-2 will be interpreted as result of Variscan shortening and crustal imbrication and underthrusting, as observed in profile ESCIN-3.3 and in the upper crust.

> The points made above indicate that I found the rolling discussions on the tectonic interpretation rather confusing. This may reflect the difficulties in reconciling competing views amongst the extensive authorship!

The first author is the only contributor that has been part of all the experiments and most publications regarding the presented extensive dataset. It is her model which is presented here, with the agreement of co-authors. However, we will try to clarify the tectonic interpretation

> The Geological Setting notes are useful but quite involved, detailed and dense. The only illustration that accompanies the text is the geo-tectonic map of Iberia. As such it is very difficult to follow. How much of this do I, as a reader, need to retain to pick up the story...? For example, is the timing and delay of anatexis (line 132) really needed for the interpretation of the seismic data later?

We think that the Geological setting contains what is necessary to understand our interpretation, and yes, there are way too many things to retain. Many of the data described there will find support in the geological sections that we are going to add to each profile. In addition, we consider that timing of late Variscan extension (and anatexis) is important as it overprints pre-and early Variscan features and erases, in places, the mid-crustal detachment, with the corresponding consequences.

> The message I get from the "Geological Setting" is that the structure of the present-day near surface is complex. . . including folds – that include deformed thrusts and thrust sheets (e.g. lines 150-157; line 187) – which is not conducive to their seismic imaging. . . For readers not familiar with region, some kind of palaeotectonic framework diagram could help to reinforcing the content. Likewise, some simple diagrams illustrating the competing models and interpretations of crustal structure would be useful – and these could then allow the seismic interpretations of the composite profile to be reframed as tests against these models.

We agree with the reviewer that geology is extremely complex. But entering on paleo-tectonic (plate-dynamics?) considerations is outside the scope of the paper and will make it even more complicated. Our goal is to make a joint interpretation of the entire dataset, understanding the role of observed reflectors in the context of the Iberian Massif, which implies the collision between Gondwana and Laurussia (Avalonia). Accordingly, geological cross-sections are going to be added to all seismic profiles.

> Line 770 etc alludes to important ambiguities resulting from the interpretation of out-ofplane and migration artifacts. More could be made of this in discussion of interpretation uncertainties.

Comments on the uncertainties of the interpretation on the edge of the profiles and in areas with 3D reflectivity will be added

> The interpretation is interwoven with basic description of seismic character. I think the narrative would flow better if the seismic reflector patterns were described first and then interpreted. The

narrative would benefit from a simple statement of assumptions and the preferred model at the outset (see above) – as much of the discussion here takes much of this as read.

As indicated above, we prefer to keep this structure as it is more agile, but we are going to add extra information in the figure captions to help in the interpretation

For example – line 461 and on states that the variations in the thickness (in TWT) of the reflective layer ("lower crust") imply differential thinning– extension: But why? Could it not be that the reflectivity was developed heterogeneously? Or that the thicker portions have been thickened, rather than the thinner ones thinned?

The thinned portions of the lower crust appear in the areas where late Variscan extension has been described by metamorphic offsets, existence of gneiss domes and pervasive crustal melting. Where these processes have not been identified, the lower crust is thicker. In our opinion, these facts allow to make the interpretation we have presented

Section 3.3 Is called a description of the seismic sections. It would be better indeed if this was what it was.. In fact, the section interlaces basic description of the seismic character with geological interpretation. In my view, the narrative would flow better if these two aspects were decoupled – so that first order description ("observations") are separated from the interpretation. So describe reflection dips…Then say you infer that these track shear zone/thrust zone trajectories. Therefore where they go sub-horizontal then you deduce regional floor thrust positions

We hope the reviewer allows us to keep the present structure, adding information on the figure captions and cross-sections on top of the seismic profiles to ease the interpretation. Also, we will change the name of section 3.3. The reasons to do this are explained above.

Section 4.3 There are not many places in the world, away from Cenozoic orogens and basins, where continental crust is not underlain by a largely sub-horizontal Moho. Whether this represents gravitation flow of deep crust or simply differential isostatic rebound and concomitant erosion is debatable. Just how much upper crustal extension is there (stretching factors) from place to place?

Extension is the late tectonic dominant process in areas where high degree rocks crop out (migmatites and granites), i.e, NW Iberian Massif. Isostatic balance must be the dominant process elsewhere (S-SW Iberian Massif). We don't have absolute control about extension since crustal melting has erased much of this information. But in places, metamorphic offsets related to extension have been estimated (e.g., chlorite to sillimanite in Central Iberia; Díez Balda et al., 1995). The minimum offset of some normal faults and extensional detachments has also been estimated (e.g., Viveiro fault, 14 km by thermobarometry, chlorite zone to sillimanite zone: e.g., Martínez et al., 1996). But these zones do not necessarily coincide with locations sampled by the seismic profiles. Nevertheless, we will include these data.

In settings like the Variscan – is the Moho a passive pre-orogenic marker – or is it a (partly magmatic or metamorphic) construct? There are interesting points in this discussions – many further references could be added

In our opinion the Moho is a modified pre-orogenic marker and a detachment along most of the profile. However, it has been redefined in areas of severe extension (ESCIN3-2), and where Alpine reworking has affected the lower crust (CIMDEF). That implies that along the Variscan Massif the Moho has a varying character depending on the tectonic evolution.

Section 4.4 I found the premise here confusing. Metamorphic units are notoriously metastable – after all we get granulites and eclogites at the Earth's surface. Only if the metamorphism was in equilibrium and therefore over-printed previous assemblage salong modern (sub-horizontal) isotherms would the crustal seismic structure be as discussed here. But if so -the tectonic structure would (presumably) be hard to resolve– the intensities of reflectivity in the profiles could simply chart metamorphic (thermal) structure – not intensities of deformation as assumed here). : You allude to this (line 761-2). But if so – when is the layering established? Presumably post-tectonically (after thermal re-equilibration)

The discussion is intended to show that a petrological subdivision of the crust would yield an extremely complex crustal section, as the same rocks may have passed through the upper, middle and even lower crust and then be back to upper crustal levels. Conversely, a subdivision can be made on the basis of seismic profiling which reflects the present state of the crust and is (perhaps surprisingly) much simpler as depends on crustal rheology at the timing of deformation. This is the goal of the paper.

In the final discussion on the mid-crustal structure – description of the geophysical character is continuously intermingled with interpretation as a tectonic discontinuity. I would find it helpful if these two distinct aspects were separated. By all means set up the discussion in terms of Conrad – which is a geophysical construct. But make this distinct from its geo-tectonic interpretation.

We will make the necessary changes to separate these two points in the discussion

Personally, I find the continual use of acronyms distracting – especially short ones. It is easier for readers if you use Cantabrian Zone rather than CZ for example.

We will make an acronyms list where the reader can refer to easily

Line 57 – more complete than what? Better to say Our aim here is the present a composite seismic profile that integrates results from two new experiments (CIMDEF and ALCUDIA WA) with existing data-sets (specify).

We'll do it

Line 60 – "Later on" makes it sound like it is another paper. "Here we continue to..."or some such might be clearer...continuing...We revisit interpretations of crustal extension and a possible mid-crustal discontinuity. We discuss mid-crustal reflectivity, the so-called "Conrad Discontinuity" of classical continental seismology (Conrad 1921), in the light of long-running debates as to its tectonic significance (REFS).

We'll change it

Line 87 – strictly the correlation does "support" the affinity – it is consistent with it ..

We'll change it

Line 88 etc "Evidences" - the plural of evidence in this context is "evidence" (no "s", like sheep).

We'll change it

Line 95 – "in the surface" – do you mean at outcrop?

Yes, we will rephrase it

Line 98 – what is "it"? The structure of the Iberian Massif along a N-S transect?

The diachronous deformation of the Iberian Massif

Line 139 and Line 140 etc. Be consistent with the verb….– is it "crop out" or "outcropping"..

According to reviewer 1, we will change it to outcropping

Line 183-184. Statements like this are key….mid crustal reflectivity can be explained by intrusions: But what evidence is there that they were controlled by shear zones?

The relation between reflectivity, intrusions and shear zones has been brought up by Schmelzbach et al., 2007 and mostly 2008 (line 494) and it is supported by surface geology (Simancas et al., 2003).

Why does reflectivity necessarily track deformation?

Because it follows the pattern of faults and thrusts in cross sections. But event though reflectivity follows the geometry of these features, it is probably due to the fact that thrusts often follow the boundaries between lithologies. The later will be added on the top of the seismic sections

Line 229 etc. a plural of a date has no apostrophe – it's 1990s

We'll change it

Line 237 – kind of experiment (no need for plural).

We'll change it

Line 285-287. Please reference explicitly these primary sources for the seismic processing. Hopefully these are peer-reviewed, formal publications!

We will do it

Line 305 (and many other places). Interpretation is presented as fact. So "W-dipping reflections that represent the Variscan imbrication" – is highly interpretative. First it would help if this statement is justified…. How explicitly does the reflectivity match to outcrop structure?

In relation to this point, the addition of cross sections in the top of seismic profiles will ease the interpretation

Line 312 – Interesting – but when thin-skinned interpretations were provided by (eg) COCORP Appalachians from 1970s– they tied reflectivity to underthrust sediments that could be traced down from outcrop….

Yes, often thrusts follow lithological boundaries that feature a high impedance contrast. This point will be considered throughout the text

Line 448 etc. I'd avoid using the phase "is related to" when discussing the seismic expression with respect to the surface geology. A better basic phrase is – "coincides with" – as this avoids associating description with interpreted causation….

We'll do it

Line 459 – Can you exclude the "cross-cutting" relationships are in-plane migration (or out of plane) artifacts?

They have the geometry of Variscan structures and since the profile is perpendicular to the latter, we think they are in plane reflections. In addition, they migrate as they are expected for 2D structures.

Line 477 "Mantle" reflectivity – what evidence is there to support the notion of crust mantle imbrication? Could this not be intra-mantle structure?

This reflection is related to a crocodile structure well identified in the lower crust. The most likely explanation is that the reflectivity into the mantle is part of this structure, i.e., crust imbricated into the mantle. The geological context does not suggest the existence of a shear zone in the mantle, which, in addition, might not have this high reflectivity.

Line 707 – which author? Meissner??

Yes, Meissner, 1989. We will change it to 'the later author'

---

## Author Response (AR2)

Response to reviews on the MS: **"Evolution of the Iberian Massif as deduced from its crustal thickness and geometry of a mid-crustal (Conrad) discontinuity"**, by Ayarza et al.

**Reviewer 1 and editor**

Dear editor,

Suggested revisions have been carried out. The Geological Setting has been shortened and organized in subsections. Non relevant parts of the text have been removed. The Discussion has been also slightly shortened and also some parts have been divided in subsections. Finally, English Grammar has been further revised throughout the text and the paper has been accordingly edited.

We understand this is a long paper, probably complex for those not familiar with the Iberian Massif, as it presents a compendium and re-interpretation of all normal incidence seismic datasets. However, we consider it very appropriated for this special volume and we are happy to include it there.

Best regards